# Standard multiscale entropy reflects neural dynamics at mismatched temporal scales: What's signal irregularity got to do with it?

**Julian Q. Kosciessa**[1,2,3]*, **Niels A. Kloosterman**[1,2], **Douglas D. Garrett**[1,2]*

**1** Max Planck UCL Centre for Computational Psychiatry and Ageing Research, Berlin, Germany, **2** Center for Lifespan Psychology, Max Planck Institute for Human Development, Berlin, Germany, **3** Department of Psychology, Humboldt-Universität zu Berlin, Berlin, Germany

* kosciessa@mpib-berlin.mpg.de (JQK); garrett@mpib-berlin.mpg.de (DDG)

**Data Availability Statement:** Raw empirical data is provided at https://osf.io/q3vxm/ (DOI 10.17605/OSF.IO/Q3VXM). Code used to produce simulations, empirical analyses and figures is

## Abstract

Multiscale Entropy (MSE) is used to characterize the temporal irregularity of neural time series patterns. Due to its' presumed sensitivity to non-linear signal characteristics, MSE is typically considered a complementary measure of brain dynamics to signal variance and spectral power. However, the divergence between these measures is often unclear in application. Furthermore, it is commonly assumed (yet sparingly verified) that entropy estimated at specific time scales reflects signal irregularity at those precise time scales of brain function. We argue that such assumptions are not tenable. Using simulated and empirical electroencephalogram (EEG) data from 47 younger and 52 older adults, we indicate strong and previously underappreciated associations between MSE and spectral power, and highlight how these links preclude traditional interpretations of MSE time scales. Specifically, we show that the typical definition of temporal patterns via "similarity bounds" biases coarse MSE scales–that are thought to reflect slow dynamics–by high-frequency dynamics. Moreover, we demonstrate that entropy at fine time scales–presumed to indicate fast dynamics– is highly sensitive to broadband spectral power, a measure dominated by low-frequency contributions. Jointly, these issues produce counterintuitive reflections of frequency-specific content on MSE time scales. We emphasize the resulting inferential problems in a conceptual replication of cross-sectional age differences at rest, in which scale-specific entropy age effects could be explained by spectral power differences at mismatched temporal scales. Furthermore, we demonstrate how such problems may be alleviated, resulting in the indication of scale-specific age differences in rhythmic irregularity. By controlling for narrowband contributions, we indicate that spontaneous alpha rhythms during eyes open rest transiently reduce broadband signal irregularity. Finally, we recommend best practices that may better permit a valid estimation and interpretation of neural signal irregularity at time scales of interest.

provided at https://git.mpib-berlin.mpg.de/LNDG/
rhythms_entropy. The code implementing the
mMSE algorithm is available from https://github.
com/LNDG/mMSE.

**Funding:** This study was conducted within the
'Lifespan Neural Dynamics Group' within the Max
Planck UCL Centre for Computational Psychiatry
and Ageing Research in the Max Planck Institute
for Human Development (MPIB) in Berlin,
Germany. DDG and NAK were supported by an
Emmy Noether Programme grant (to DDG) from
the German Research Foundation, and by the Max
Planck UCL Centre for Computational Psychiatry
and Ageing Research. JQK is a pre-doctoral fellow
supported by the International Max Planck
Research School on Computational Methods in
Psychiatry and Ageing Research (IMPRS
COMP2PSYCH). The participating institutions are
the Max Planck Institute for Human Development,
Berlin, Germany, and University College London,
London, UK. For more information, see https://
www.mps-ucl-centre.mpg.de/en/comp2psych. The
funders had no role in study design, data collection
and analysis, decision to publish, or preparation of
the manuscript.

**Competing interests:** The authors have declared
that no competing interests exist.

## Author summary

Brain signals exhibit a wealth of dynamic patterns that are thought to reflect ongoing neural computations. Multiscale sample entropy (MSE) intends to describe the temporal irregularity of such patterns at multiple time scales of brain function. However, the notion of time scales may often be unintuitive. In particular, traditional implementations of MSE are sensitive to slow fluctuations at fine time scales, and fast dynamics at coarse time scales. This conceptual divergence is often overlooked and may lead to difficulties in establishing the unique contribution of MSE to effects of interest over more established spectral power. Using simulations and empirical data, we highlight these issues and provide evidence for their relevance for valid practical inferences. We further highlight that standard MSE and traditional spectral power are highly collinear in our example. Finally, our analyses indicate that spectral filtering can be used to estimate temporal signal irregularity at matching and intuitive time scales. To guide future studies, we make multiple recommendations based on our observations. We believe that following these suggestions may advance our understanding of the unique contributions of neural signal irregularity to neural and cognitive function across the lifespan.

## Introduction

### Entropy as a measure of signal irregularity

Neural times series exhibit a wealth of dynamic patterns that are thought to reflect ongoing neural computations. While some of these patterns consist of stereotypical deflections [e.g., periodic neural rhythms; 1, 2], the framework of nonlinear dynamics and complex systems also emphasizes the importance of temporal irregularity (or variability) for healthy, efficient, and flexible neural function [3–6]. Specifically, functional network dynamics may reflect the non-linear interaction of local and global population activity, for which intermediate levels of network noise theoretically afford high network capacity and dynamic range [7–10]. In parallel with such conceptual advances, multiscale entropy (MSE) [11, 12], an information-theoretic index that estimates sample entropy [13] at multiple time scales (Fig 1A), has become a promising tool to quantify the irregularity of neural time series across different brain states, the lifespan, and in relation to health and disease [14–22]. However, we argue that outstanding methodological issues regarding the mapping of neural-to-MSE time scales reduce the current interpretability of MSE results, and–if not properly accounted for–limit MSE's utility for investigating substantive neurocomputational questions of interest.

   In general, sample entropy quantifies the irregularity of temporal patterns in a given signal (for an example of its calculation, see Fig 1B). Whereas signals with a repetitive structure (like stationary signals or rhythmic fluctuations) are estimated as having low entropy, less predictable (or random) signals are ascribed high entropy. As an extension of this principle, MSE aims to describe temporal irregularity at different time scales–varying from fine (also referred to as 'short') to coarse (or 'long'). In conventional Fourier analysis of time series data, time scales are quantified in terms of lower and higher frequencies present in the signal. This has been shown to be a principled time scale descriptor that relates at least in part to structural properties of the generating neural circuits [2, 23–26]. Given this meaningful definition of fast and slow events, it is a common assumption–including in guides to MSE's interpretation in neural applications [27]–that fine-to-coarse scales characterize the irregularity of high-to-low frequency dynamics, respectively. However, here we highlight one methodological and one

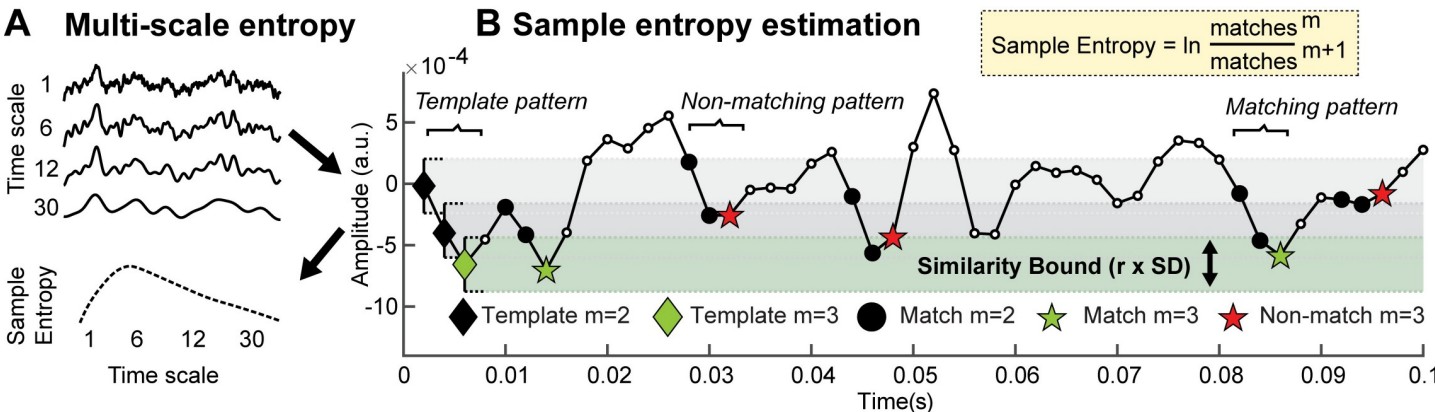

**Fig 1. Traditional MSE estimation procedure.** (**A**) Multi-scale entropy is an extension of sample entropy, an information-theoretic metric intended to describe the temporal irregularity of time series data. To estimate entropy for different time scales, the original signal is traditionally 'coarse-grained' using low-pass filters, followed by the calculation of the sample entropy. (**B**) Sample entropy estimation procedure. Sample entropy measures the conditional probability that two amplitude patterns of sequence length m (here, 2) remain similar (or matching) when the next sample *m* + 1 is included in the sequence. Hence, sample entropy increases with temporal irregularity, i.e., with the number of m-length patterns that do not remain similar at length m+1 (non-matches). To discretize temporal patterns from continuous amplitudes, similarity bounds (defined as a proportion *r*, here .5, of the signal's standard deviation [SD]) define amplitude ranges around each sample in a given template sequence, within which matching samples are identified in the rest of the time series. These are indicated by horizontal grey and green bars around the first three template samples. This procedure is applied to each template sequence in time, and the pattern counts are summed to estimate the signal's entropy. The exemplary time series is a selected empirical EEG signal that was 40-Hz high-pass filtered with a 6th order Butterworth filter.

conceptual issue regarding the computation of MSE that challenge such a direct scale-to-frequency mapping. First, we show that the traditional definition of temporal patterns may lead to an influence of high frequencies on coarse entropy time scales (Issue 1). Second, we highlight that the signal content at fine time scales renders entropy estimates sensitive to a conjunction of scale-free and narrowband signals, including slow fluctuations (Issue 2).

Due to its assessment of temporal patterns rather than sinusoidal oscillatory dynamics, MSE has been motivated as a complementary measure to spectral variance/power that is sensitive to multi-scale, potentially non-linear, signal characteristics, such as phase shifts or cross-frequency coupling. [Note that we use the terms power and variance interchangeably, as a time domain signal's broadband variance is proportional to the integral of its power spectral density, while narrowband variance in the time domain is identical to narrowband power in the spectral domain.] However, the overlap between these measures is often unclear in application because the mapping between spectral power and scale-wise entropy is ambiguous. Such ambiguity affects both the ability to compare individuals at any scale, and the ability to compare entropy levels across scales within person. We argue that a clarification of these issues is thus necessary for valid inferences of time scale-specific 'neural irregularity' in a growing number of neuroscientific MSE applications.

### Issue 1: Global similarity bounds introduce a scale-dependent variance bias

A principle assumption of sample entropy is that "the degree of irregularity of a complex signal [. . .] cannot be entirely captured by the SD [i.e., standard deviation]" [28; i.e., square root of variance]. To ensure this, sample entropy is typically assessed relative to the standard deviation of the broadband signal to intuitively normalize the estimation of irregularity for overall distributional width [13, 14, see also 28]. In particular, the ***similarity bound***–defined by a constant r, by which the signal SD is multiplied–reflects the tolerance for labeling time points as being similar or different, and thus, determines how liberal the algorithm is towards detecting 'matching patterns' (Fig 2A and 2B). While wider bounds decrease entropy estimates,

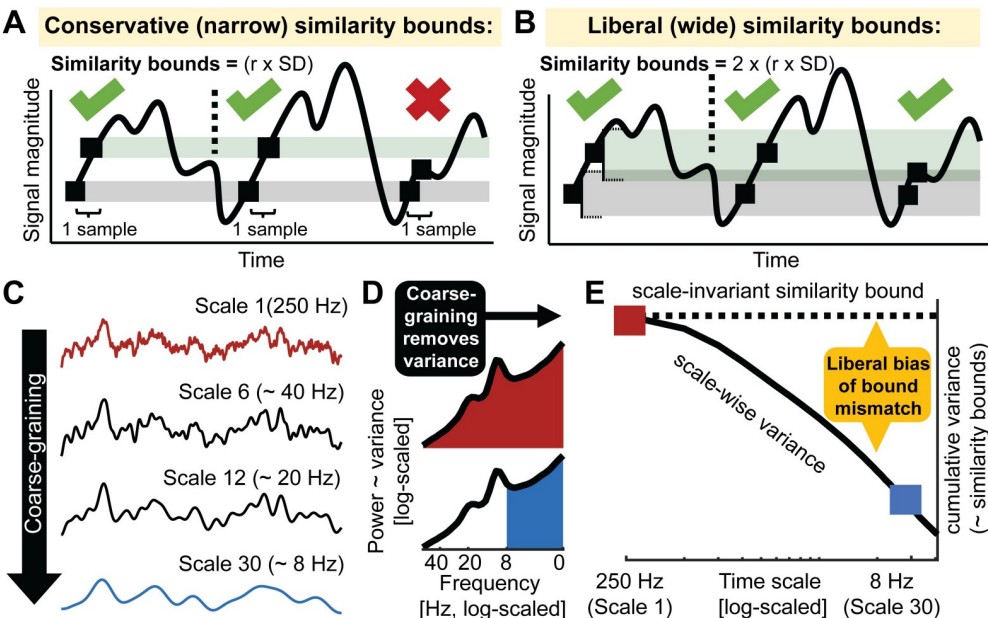

**Fig 2. Issue 1: Global similarity bounds systematically confound the entropy of coarse-scale signals with removed spectral power.** (**A, B**) Similarity bounds constrain sample entropy as shown schematically for entropy estimation using narrower (**A**) and wider (**B**) similarity bounds. For clarity, only a subset of pattern matches (green ticks) and mismatches (red cross) are indicated for a sequence length m = 1(cf. Fig 1B). Wider, more liberal similarity bounds indicate more pattern matches than narrow, conservative bounds, thereby decreasing entropy. S2 Fig shows the empirical link between liberal similarity bounds and sample entropy estimates. (**C-E**) Divergence between global similarity bounds and scale-wise signal SD biases coarse-scale entropy. (**C**) Coarse-graining (see Fig 1A) progressively reduces variance from the original broadband signal (as shown in panel E). (**D**) At original sampling rates (i.e., time scale 1; marked red in panels DE and F), neural signal variance is usually composed of broadband 1/f content and narrowband rhythmic peaks. Note that the x-axis plots decreasing frequencies to align with the traditional MSE low-pass filter direction. Towards coarser scales (e.g., scale 30; marked blue in CD and E), signal variance progressively decreases, as the signal becomes more specific to low frequencies. (**E**) Due to the systematic and cumulative reduction of variance in scale-wise signals, global similarity bounds become liberally biased ('broad'). Critically, systematic differences in the magnitude of this bias (e.g., due to different spectral slopes) introduce systematic entropy differences at coarser scales.

narrower bounds increase them [13, 29, 30] (S2 Fig). Crucially, the similarity bound is often not equally liberal across time scales, resulting in an entropy estimation bias. Specifically, to characterize temporal irregularity at coarser time scales, signals are typically successively low-pass filtered [or 'coarse-grained'; 31] (Fig 2C), whereas the similarity bound typically (in its 'Original' implementation) is set only once–namely relative to the SD of the original unfiltered signal. Due to the progressive filtering, coarse-graining successively reduces the signal's SD, yet a single global (i.e., scale-invariant) similarity bound remains based on the cumulative variance of all estimable frequencies (Fig 2D and 2E). As a result, the similarity bound becomes increasingly liberal towards pattern similarity at coarser scales, thereby reducing entropy estimates. This is most clearly illustrated by the observation that white noise signals, which should be characterized as equally random at each time scale, exhibit decreasing entropy values towards coarser scales when global *similarity bound*s are used [27, 29, 32]. This issue has been recognized previously [29], and provided a rationale for recomputing the *similarity bound* for each time scale [29, 33–35]. But despite the benefits of this refinement that was already proposed fifteen years ago, our review of the literature revealed that the use of global bounds remains dominant in over 90% of neuroscientific MSE applications (see S1 Text) and in previous validation work [27]. Crucially, the consequences of this bias for practical inference remain

unclear. We therefore argue that a comprehensive assessment of the resulting bias is needed to highlight this issue, both to clarify previous results and to guide future studies.

## Issue 2: Traditional scale definitions lead to diffuse time scale reflections of spectral content

While matched similarity bounds account for *total signal variation* at any specific time scale, sample entropy remains related to *the variance structure* (i.e., the power spectrum) of the signal as *one* indicator of its temporal irregularity [4]. Most neural signals exhibit a scale-free $\frac{1}{f^x}$ power distribution [36–38], for which the exponent $x$ indicates the prevalence of low-to-high-frequency components in the signal. This ratio is also referred to as the power spectral density (PSD) slope. Smaller exponents (indicating shallower PSD slopes) characterize signals with relatively strong high-frequency contributions (i.e., reduced temporal autocorrelations, and less predictability) compared to larger exponents that indicate steeper slopes. This conceptual link between PSD slopes (or high-to-low frequency power ratios that may have strong broadband slope contributions [39]) and sample entropy has been empirically observed across subjects, wakefulness and task states [14, 17, 40]. However, the sensitivity of fine-scale entropy to PSD slopes–a multi-scale characteristic–highlights that the contribution of slow-to-fast signal content to fine-scale entropy is unclear. This ambiguity arises from the algorithm that derives scale-wise signals. In particular, 'Original' MSE implementations use low-pass filters to derive signals at coarser time scales, which increasingly constrains entropy estimates to slower fluctuations. As such, each scale defines an upper bound for the range of included frequencies (see methods). However, the opposite is not true, resulting in a lack of high-frequency specificity. Hence, finer time scales characterize the *entire* broadband signal (see Fig 3A) which represents a non-specific mixture of low and high-frequency elements across scale-free and rhythmic signal contributions [41, 42]. Crucially, the contribution of these elements to neural broadband signals is not equal. Rather, the variance of $\frac{1}{f^x}$ signals is dominated by the amplitude of low frequencies, which may thus disproportionally impact the assessment of pattern irregularity [35]. As a result, broadband signal characterization challenges the assumption that fine-scale entropy mainly describes 'fast' events. More generally, this highlights large uncertainty regarding the frequencies that are represented at *any* particular time scale.

The projection of narrowband rhythms into simulated noise signals provides a well-controlled situation in which to study the mapping of neural irregularity to MSE, due to their clearly defined time scale (i.e., period = inverse of frequency) and regularity (added rhythmic variance = more regular signal = decreased entropy). Moreover, rhythmic structure remains a dominant target signal in neuroscience [1, 36, 43] for which entropy, as a complementary descriptor, should provide an anti-correlated reflection. However, previous simulations on the mapping of rhythms onto MSE time scales have produced puzzling results that have received little attention in the literature so far; while a linear mapping between rhythmic frequency and entropy time scales has been observed, added rhythmic regularity has been shown to *increase* entropy above baseline in previous work [4, 22, 44]. This notably contrasts with the intuition that added signal regularity should reduce observed entropy. Thus, additional simulations are necessary to assess the intuitive notion that rhythmicity should be anticorrelated with entropy, and to investigate whether this phenomenon indeed occurs at specific time scales, as previously assumed [4, 22, 44]. In particular, we probed the feasibility of using high-pass and band-pass filters (relative to standard low-pass options) to control the MSE time scales at which rhythmicity would be reflected (Fig 3B).

In summary, Issue 1 suggests a coarse-scale bias introduced by global similarity bounds, and Issue 2 highlights a mixture of narrow- and broadband contributions to fine scales. In

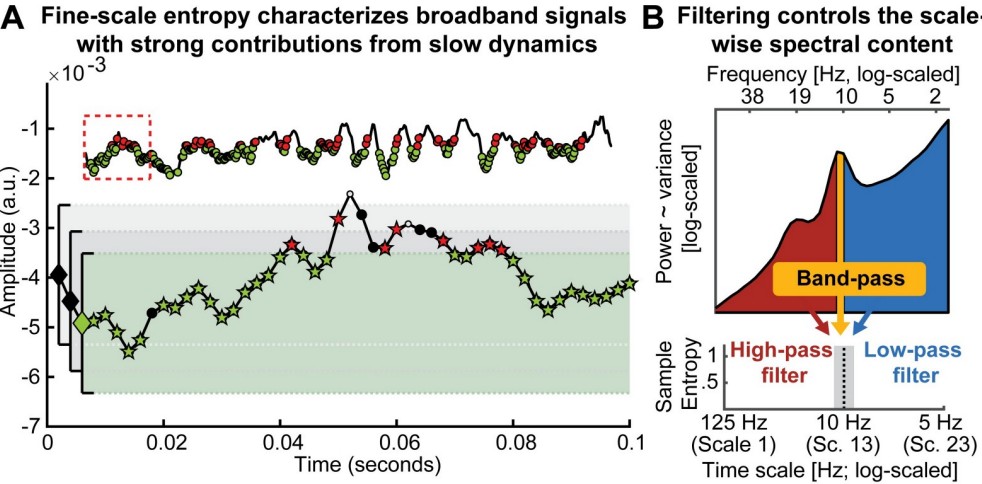

**Fig 3. Issue 2: Traditional scale derivation leads to diffuse time-scale reflections of spectral power.** (A) Exemplary sample entropy estimation in the same empirical EEG signal shown in Fig 1B, but without application of a high-pass filter, thus including dominant slow dynamics. See Fig 1B for a legend of the Figure elements. In brief, green elements indicate pattern matches at m+1, whereas red elements indicate pattern mismatches at m+1. In the presence of large low-frequency fluctuations, sample entropy at fine scales (here scale 1) may to a large extent characterize the temporal regularity of slow dynamics. Note that this is not a case of biased similarity bounds, but a desired adjustment to the large amplitude of slow fluctuations. The inset shows an extended segment (800 ms) of the same signal, allowing for an assessment of the slower signal dynamics. The red box indicates the 100 ms signal shown in the main plot. (B) A scale-wise filter implementation controls the scale-wise spectral content, as schematically shown here for the filter-dependent representation of spectral content at a time scale of approximately 10 Hz (for a note on the x-axis labeling, see methods: *Calculation of multi-scale sample entropy*). Traditionally, low-pass filters are used to derive coarser scales, which introduces a sensitivity to slower fluctuations. However, other filter implementations can be used to e.g., investigate the pattern irregularity of fast signal variations. No matter whether low or high pass filters are used, the spectral content influencing entropy estimates is by definition not specific to any particular time scale; band-pass filters provide one viable solution permitting such specificity.

worst-case scenarios, a conjunction of these issues may lead to a reflection of fast dynamics in coarse entropy and a reflection of slow dynamics in fine entropy, thus paradoxically *inverting* the intuitive time scale interpretation. These issues have not been jointly assessed, however, and there is little evidence of whether and how these methodological issues may impact practical inferences motivated by neurobiological questions of interest. We focus on two example scenarios in the current study.

## Impact of issues on practical inferences: (1) age differences in neural irregularity at fast and slow time scales

One principal application of multiscale entropy is in the domain of lifespan covariations between neural dynamics and structural brain network ontogeny [for a review see 45]. Within this line of inquiry, it has been proposed that structural brain alterations across the lifespan manifest as entropy differences at distinct time scales [16, 18, 40, 46]. Specifically, it has been suggested that coarse-scale entropy decreases and fine-scale entropy rises with increasing adult age as a reflection of senescent shifts from global to increasingly local information processing [16, 18]. Crucially, this mirrors observations based on spectral power, where age-related decreases in the magnitude of low-frequencies [47, 48] are accompanied by increases in high-frequency activity, conceptualized also as a flattening of power spectral density (PSD) slopes [16, 18, 40, 49]. These results seemingly converge towards a joint decrease of low-frequency power and coarse-scale entropy in older adults (and an increase for both regarding fast dynamics). However, this correspondence is surprising upon closer inspection given the

presumed anticorrelation between the magnitude of signal regularity (as indicated by heightened spectral power) and entropy. In light of concerns regarding the interpretation of entropy time scales (see above), we assessed cross-sectional age effects on both MSE and spectral power as a test case for potential mismatches in scale-dependent inferences.

### Impact of issues on practical inferences: (2) narrowband modulations of broadband irregularity

Identifying the time scale contributors to MSE is further relevant due to the assumed functional separability of narrow- and broadband brain dynamics. Whereas narrowband rhythms have been closely associated with synchronous population spiking at the service of temporal information coordination [50], scale-free broadband dynamics may provide a complementary index of the level of neocortical activation and aggregate spiking activity in humans [38, 51–53]. In particular, shallower PSD slopes have been proposed as a signature of enhanced cortical excitability (or 'neural noise') [54]. Such excitability in turn may regulate the available range of network dynamics as reflected in information entropy [10]. Notably, interactions between narrow- and broadband activity are neurobiologically expected. In particular, as the magnitude of narrowband alpha synchronization increases, population output is thought to decrease [55]. However, the methodological conflation of narrow- and broadband contributions to entropy (see "Issue 2" above) may complicate principled investigations regarding their neurobiological coupling in practice. As a corollary goal in the present work, we therefore investigate whether a principled separation of narrow- and broadband contributions to entropy is tractable.

### Current study

Here, we aimed to address two issues of frequency-to-scale mapping and their relevance for empirical applications. First, we simulated variations in rhythmic power and frequency to probe the relationship between rhythmicity and MSE time scales. Primarily, our goal was to assess how global similarity bounds (Issue 1) and the scale-wise spectral content of the analyzed signal (Issue 2) influence the time scales at which added rhythmicity is observed. Then, we attempted to replicate reported cross-sectional age differences in human electroencephalography (EEG) signals recorded during rest. We assessed whether younger adults would show increased coarse scale and decreased fine-scale entropy compared to older adults, and we probed the extent to which such scale-specific results depend on mismatched spectral power via the issues above. As corollary goals, we assessed the potential of band-pass and band-stop approaches for deriving more intuitive insights regarding the time scales of signal irregularity. First, we probed the potential of 'frequency-specific' estimates of signal irregularity via band-pass filters, and assessed age differences therein. Second, we assessed the relation between alpha rhythms and broadband signal irregularity, after accounting for their methodological coupling. We refer to traditional settings that use global bounds and low-pass filtering as 'Original' throughout the remainder of the manuscript (see methods for details).

### Results

### Simulations indicate a diffuse mapping between rhythmicity and MSE time scales as a function of global similarity bounds and spectral signal content

Our first aim was to probe how scale-specific events, namely rhythms of a given frequency, modulate MSE time scales. For this purpose, we simulated 10 Hz (alpha) rhythms of varying power on top of pink noise and calculated the MSE of those signals. First, we probed the influence of global similarity bounds (as used in 'Original' implementations) on the time scale

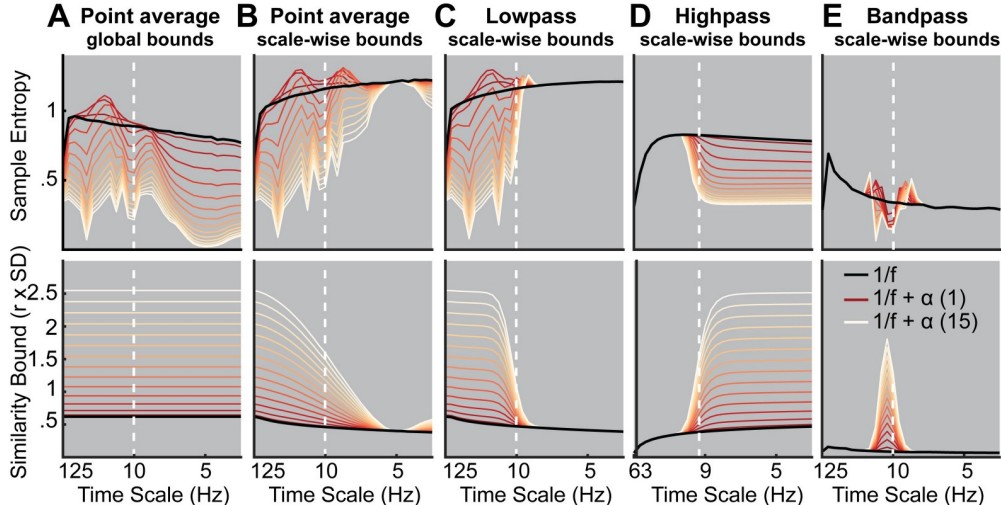

**Fig 4. Rhythmic power manifests at different time scales depending on filter choice and similarity bound.**
Simulations indicate at which time scales the addition of varying magnitudes of stereotypic narrowband 10 Hz
rhythms (red-to-white line color gradient) modulate entropy compared to the baseline 1/f signal (black line).
Simulations indicate that increases in rhythmicity strongly reduce entropy estimates alongside increases in the
similarity bound. The affected scales vary as a function of global vs. scale-dependent similarity bounds and the spectral
filtering used to derive coarser time scales. Crucially, in 'Original' implementations, added narrowband rhythmicity
decreased entropy with low scale-specificity, in line with global increases in the similarity bound (A). In contrast, the
use of scale-varying thresholds (B) and dedicated filtering (C-E) increased specificity regarding the time scales at which
rhythmicity was reflected. Note that timescales are presented in Hz to facilitate the visual assessment of rhythmic
modulation. For all versions except high pass, the scale represents the upper Nyquist bound of the embedding
dimension. For the high pass variant, the scale represents the high pass frequency (see *methods*). Time scales are log-
scaled. Spectral attenuation properties of the Butterworth filters are shown in S4 Fig.

mapping (Issue 1). Crucially, as a result of using a global similarity bound for all time scales,
strong rhythmic power decreased MSE estimates across a range of time scales, including time
scales at which added 10 Hz rhythmicity did not contribute to the scale-wise signal (Fig 4A,
upper panel). As highlighted in Issue 1, this can be explained by a general increase in the liber-
ality of bounds (Fig 4A, lower panel) that introduced a bias on coarse-scale entropy below 10
Hz. In contrast, when scale-dependent similarity bounds were used with low-pass filters (Fig
4B and 4C), strong rhythmicity systematically affected entropy only at finer time scales than
the simulated frequency (i.e., to the left of the vertical line in Fig 4C, albeit in a diffuse manner,
which we will examine next).

Second, we assessed the influence of the scale-wise filters (and hence, the spectral signal
content) on frequency-to-scale mapping (see Issue 2, Fig 3B). In particular, we expected that
low-pass filters (A-C) would lead to entropy decreases at finer time scales than the simulated
frequency, whereas high-pass filters would lead to a rhythm representation at coarser time
scales (Fig 3B). In line with these expectations, low-pass filters constrained the influence of
narrowband rhythms to finer time scales (Fig 4C). As in previous work [33], Butterworth fil-
ters (Fig 4C) improved the removal of 10 Hz rhythms at coarser time scales and produced less
aliasing compared with 'Original' point-averaging (see methods, Fig 4A and 4B), with other-
wise comparable results. Hence, low-pass filters rendered multiscale entropy sensitive to vari-
ance from low frequencies, suggesting that slow events (e.g. event-related potentials) are
reflected in a diffuse manner across time scales. In contrast, high-pass filters constrained
rhythm-induced entropy decreases to coarser time scales that included 10 Hz signal content,
hence leading to estimates of high frequency entropy that were independent of low frequency
power (Fig 4D). Finally, when band-pass filters were used (Fig 4E), rhythmicity decreased

sample entropy at the target scales (despite producing edge artifacts surrounding the time scale of rhythmicity). In sum, these analyses highlight that rhythmic power increases will diffusely and non-specifically modulate MSE time scales as a function of the coarse-graining filter choice, unless a narrowband filter is applied.

Such diffuse reflection of rhythms across MSE time scales is at odds with previous simulations suggesting a rather constrained, linear mapping between the frequency of simulated rhythms and entropy time scales [4, 22, 44]. Furthermore, those studies indicated entropy *increases* with added rhythmicity, in contrast with the marked (and expected) decreases in entropy observed here. Crucially, increased entropy relative to baseline runs counter to the idea that the addition of a stereotypic pattern should decrease rather than increase pattern irregularity. To assess whether these seemingly divergent results can be reconciled, we repeated our simulation for different frequencies. We focused on a comparatively low level of rhythmicity (amplitude level = 2; SNR ~ 1.3 (see methods); S3 Fig displays exemplary time series), for which Fig 4A–4C suggested transient entropy increases above baseline. Similar to previous reports, we observed a positive association between simulated frequencies and peak entropy time scales (Fig 5) across implementations, such that rhythms of a given frequency increased entropy at slightly finer time scales (see increases in entropy above baseline to the left of the dotted vertical lines in Fig 5A–5C). However, as shown in Fig 4A–4C, such increases were counteracted when rhythmic strength increased, while global *similarity bound*s (Fig 5A) liberally biased, and thus decreased, entropy at coarser time scales (i.e., to the right of the dotted lines in Fig 5A) independent of rhythmic strength. While the mechanistic origin of entropy increases remains unclear, previous conclusions may thus have overemphasized the scale-specificity of rhythmic influences.

In sum, our simulations highlight that the choice of similarity bound and the signal's spectral content grossly affect one's ability to interpret MSE time scales. Our frequency-resolved simulations suggest that a previously argued direct frequency-to-scale mapping is not tenable when typical estimation procedures are used. Supplementing these narrowband contributions to MSE, we report results from simulations of varying spectral slopes in S2 Text and S7 Fig.

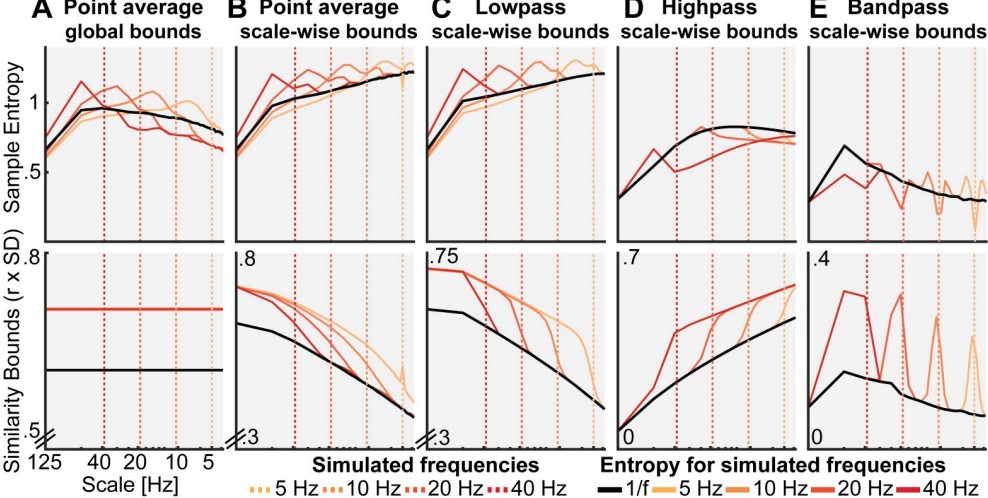

**Fig 5. Influence of rhythmic frequency on MSE estimates and similarity bounds across different MSE variants.** Simulations of different frequencies indicate a linear frequency-to-scale mapping of simulated sinusoids. Broken vertical lines indicate the simulated frequency. The original MSE variant (A) shows increased entropy at time scales finer than the simulated frequency in combination with a global entropy decrease. Low-, high- and band-pass variants exhibit the properties observed in the alpha case, with a reduction above (B, C), below (D) or at the simulated frequency (E). Time scales are log-scaled.

## Probing the impact of spectral power on MSE in a cross-sectional age comparison

Our simulations suggest profound influences of the choice of similarity bound (Issue 1) and spectral content (Issue 2) on scale-dependent MSE estimates. However, whether these issues affect inferences in empirical data remains unclear. Entropy differences across the lifespan are an important application [6], where 'Original' MSE implementations suggest that older adults exhibit higher entropy at finer time scales and lower entropy at coarser time scales compared to younger adults [for a review see 45]. Importantly, a shallowing of PSD slopes with age has also been reported, as represented by higher power at high frequencies and lower power at low frequencies [40, 49]. The raised issues of a potential (1) reflection of high frequency power on coarse scales and (2) diffuse reflection of slow spectral content thus question whether traditional MSE group differences reflect veridical differences in signal irregularity at matching time scales. Given those two issues, we specifically hypothesized that:

a. Adult age differences in coarse-scale MSE can be accounted for by group differences in high frequency power, due to the typical use of global similarity bounds (Issue 1).

b. Adult age differences in fine-scale MSE reflect differences in PSD slopes and thus depend on the contribution of low frequencies to broadband signals (Issue 2).

To assess these hypotheses, we first attempted to replicate previously reported scale-wise age differences in MSE and spectral power during eyes open rest. 'Original' settings replicated scale-dependent entropy age differences (Fig 6A1). Specifically, compared with younger adults, older adults exhibited lower entropy at coarse scales, and higher entropy at fine scales (Fig 6A1). Mirroring these results in spectral power, older adults had lower parieto-occipital alpha power and increased frontal high frequency power (Fig 6A2) compared to younger adults. This was globally associated with a shift from steeper to shallower PSD slopes with increasing age (Fig 6D). At face value, this suggests joint shifts of both power and entropy, in the same direction and at matching time scales. Crucially, however, the spatial topography of entropy differences inverted the time scale of power differences (Fig 6B and C; cf., upper and lower topographies), such that frontal high frequency power topographies resembled coarse entropy topographies (Fig 6B), while parieto-occipital age differences in slow frequency power resembled fine-scale entropy differences (Fig 6C). This rather suggests scale-mismatched associations between entropy and power.

Next, we assessed the impact of scale-wise similarity bounds and different scale-wise filters on the indication of MSE age differences (Fig 7).

Briefly, we observed three main results that deserve highlighting:

a. The implementation of scale-wise similarity bounds affected MSE age differences (Fig 7; Hypothesis A; Issue 1). In particular, with global bounds, MSE indicated increased fine-scale and decreased coarse-scale entropy for older compared to younger adults (Fig 7A1 and 7A2), in the absence of group differences in the global *similarity bound* (Fig 7A3 and 7A4). In contrast, scale-varying bounds captured age differences in variance at finer scales (Fig 7B) and abolished age differences in coarse-scale entropy (effect size was significantly reduced from r = .58 to r = .07; p = 6.8*10^-5; see Statistical analyses).

b. The chosen scale-wise filtering method also affected MSE age differences (Hypothesis B; Issue 2). Specifically, fine-scale entropy age differences were indicated when low-pass filters rendered those scales sensitive to low-frequency content (Fig 7B and 7C). Effect size did not significantly change with the adoption of scale-varying similarity bounds (from r = .44 to r = .45; p = .934). In contrast, when high-pass filters constrained fine scales to high

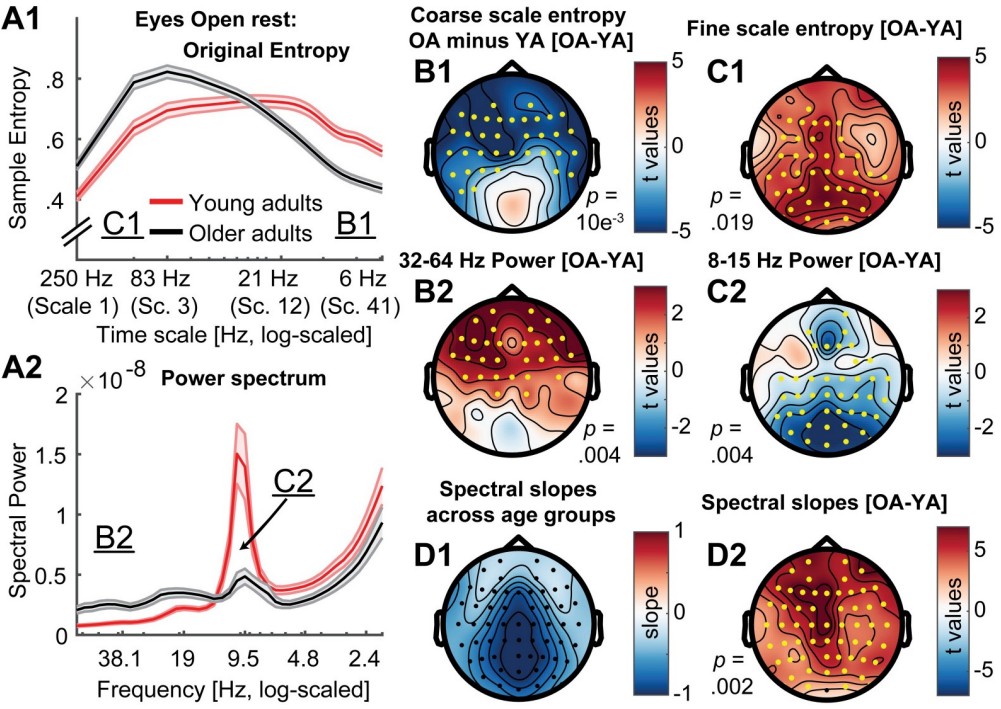

**Fig 6. Timescale-dependent age differences in spectral power and entropy during eyes open rest.** (A) MSE (A1) and power (A2) spectra for the two age groups. Error bars show standard errors of the mean. Note that in contrast to standard presentations of power, the log-scaled x-axis in A2 is sorted by decreasing frequency to enable a better visual comparison with entropy time scales (see also Fig 2D). Similarly, the x-axis in A1 has been log-scaled to allow easier visual comparison with log-scaled values in A2 and emphasize fine-scale differences (cf. Fig 7A1). Inset labels refer to the approximate time scales across which topographies are plotted in B & C. T-values of power age contrast are shown in S5 Fig. (B, C) Topographies of age differences indicate mirrored age differences in fast entropy and low frequency power, as well as coarse entropy and high frequency power. Significant differences are indicated by yellow dots. P-values correspond to the two/sided significance test of the cluster-level statistic. (D1) Spectral slopes across age groups. (D2) Age differences in spectral slopes.

frequency signals (Fig 7D), no fine-scale age differences were observed and the age effect was significantly reduced to r = .09 (p = .008).

c. Strikingly, the implementation of narrowband filters (Fig 7E) indicated two unique age effects not recoverable using other approaches: larger 'narrowband' alpha-band entropy and lower beta-band entropy for older adults compared with younger adults.

In the following sections, we assess these results more closely.

## Global similarity bounds bias coarse-scale entropy to reflect high-frequency power

Scale-dependent entropy effects in the face of global similarity bounds (as observed in the 'Original' implementation; Fig 7A) may intuitively suggest scale-specific variations in signal irregularity in the absence of variance differences. However, global similarity bounds increasingly diverge from the scale-wise signal variance towards coarser scales (Issue 1; Fig 8A). This introduces a liberal bias that systematically varies as a function of the removed variance, thereby rendering coarse MSE scales sensitive to differences in higher frequency power (i.e., Issue 1), as observed in the case of aging (Fig 8A and 8B).

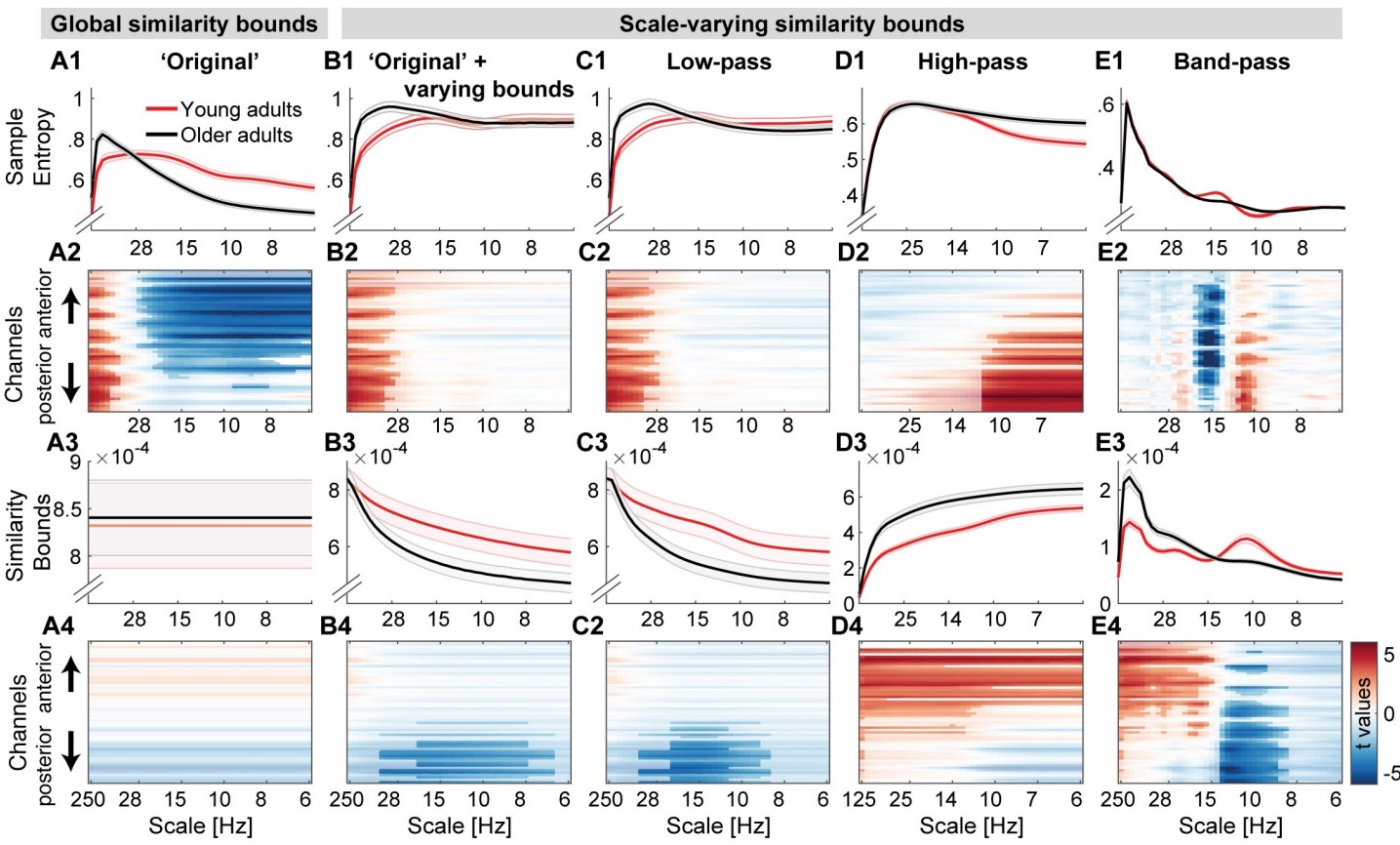

**Fig 7. Multiscale entropy age differences depend on the specifics of the estimation method.** Grand average traces of entropy (1[st] row) and similarity bounds (3[rd] row) alongside t-maps from statistical contrasts of age group differences (2[nd] + 4[th] row: younger minus older adults for entropy and bounds, respectively), shown by channel on the y-axis. Age differences were assessed by means of cluster-based permutation tests and are indicated via opacity. Original MSE (A) replicated reported scale-dependent age differences, with older adults exhibiting higher entropy at fine scales and lower entropy at coarse scales, compared with younger adults. The coarse-scale difference was exclusively observed when using global similarity bounds, whereas the fine-scale age difference was indicated with all low-pass versions (A, B, C), but not when signals were constrained to high-frequency or narrow-band ranges (D, E). In contrast, narrowband MSE indicated inverted age differences within the alpha and beta band (E).

To assess whether global bounds introduced an association between high frequency power and coarse scale entropy in the case of aging, we probed changes in *similarity bounds* and MSE between the use of global and scale-varying bounds. As expected, we observed a strong anti-correlation between inter-individual changes in *similarity bounds* and MSE (Fig 8C). That is, the more similarity bounds were re-adjusted to match the scale-wise variance, the more entropy estimates increased. Crucially, this difference was more pronounced for older adults (paired t-test; r: p = 5e-6; MSE: p = 3e-4). Due to their increased high frequency power, coarse-graining decreased older adults' scale-wise variance more so than younger adults' variance. Thus, global similarity bounds presented a more liberal threshold at coarser scales for older adults than for younger adults, in turn producing lower MSE estimates. In line with this assumed link between high frequency power and coarse scale entropy as a function of global bounds, individual high frequency power at frontal channels was anticorrelated with coarse-scale entropy estimates when a global similarity bound was applied (Fig 8D), but was dramatically weaker when the similarity bound was recomputed for each scale (YA: r = -0.15; p = .302; OA: r = .20, p = .146). This is in line with our observation that coarse-scale age differences (Fig 7A) were not found when scale-wise bounds were used (Fig 7B).

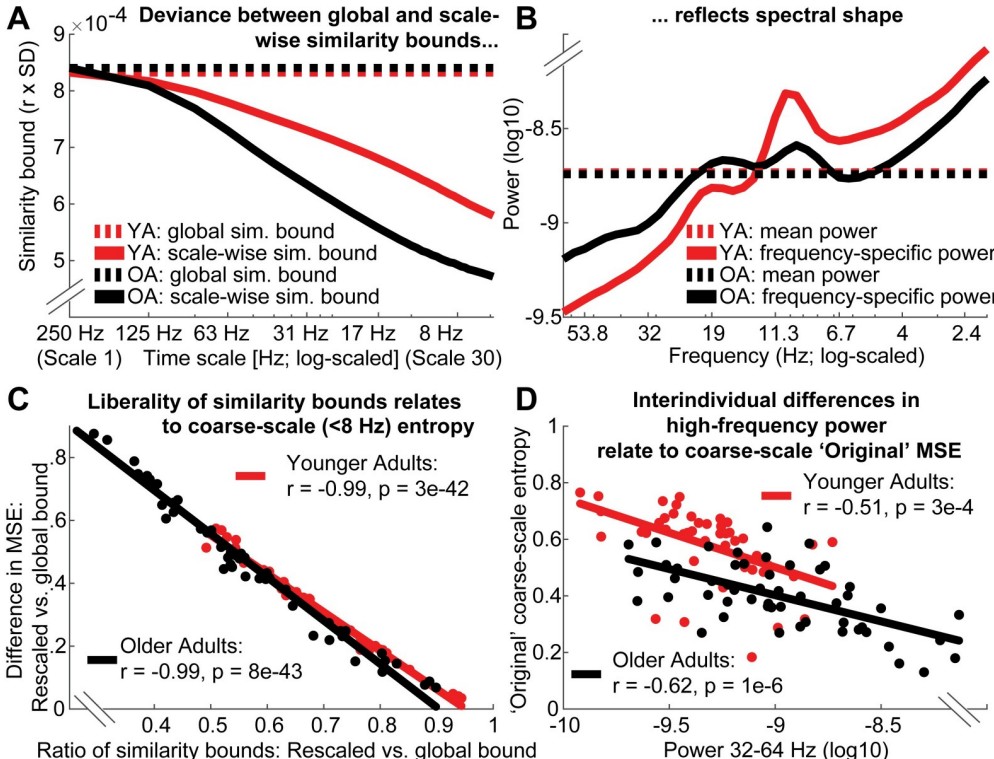

**Fig 8. Divergence of scale-specific signal variance from global similarity bounds accounts for age differences in coarse-scale entropy.** (A, B) A global similarity bound does not reflect the spectral shape, thus leading to disproportionally liberal criteria at coarse scales following the successive removal of high-frequency variance (see Fig 2C–2E for the schematic example). Scale-dependent variance is more quickly reduced in older compared to younger adults (A) due to the removal of more prevalent high-frequency variance in the older group (B). This leads to a differential bias across age groups, as reflected in the differentially mismatched distance between global and scale-dependent similarity bounds at coarser scales. (C) Removing this bias by adjusting the similarity bounds to the scale-dependent signal is associated with increases in coarse-scale entropy. This shift is more pronounced in older adults following the removal of a more prevalent bias. (D) With global similarity bounds, coarse-scale entropy strongly reflects high frequency power due to the proportionally more liberal similarity threshold associated. Low frequency power < 8 Hz was not consistently related to coarse-scale entropy (log10-power as in D; *YA: r = .12; p = .419; OA: r = .36, p = .009*). Data in A and B are global averages, data in C and D are averages from frontal 'Original' effect cluster (see Fig 7A) at entropy time scales below 8 Hz.

Taken together, these results indicate that increased high frequency power with age can account for entropy decreases at coarse time scales, whereas the pattern irregularity of slow dynamics *per se* was not modulated by age.

## Low-frequency contributions render fine-scale entropy a proxy measure of PSD slope

A common observation in the MSE literature is that MSE is highly sensitive to task and behavioral differences at fine time scales, which are assumed to reflect fast dynamics. This is surprising given that high-frequency activity remains challenging to measure [56]. Moreover, previous studies suggest that fine-scale entropy reflects power spectral density (PSD) slopes [e.g., 14, 40]. Given that 'Original' MSE implementations contain both high- and low-frequency components due to the assessment of broadband signals, we probed whether fine-scale associations with PSD slopes depend on the presence of slow fluctuations and whether age-related slope variations can account for fine-scale entropy age differences (Hypothesis B).

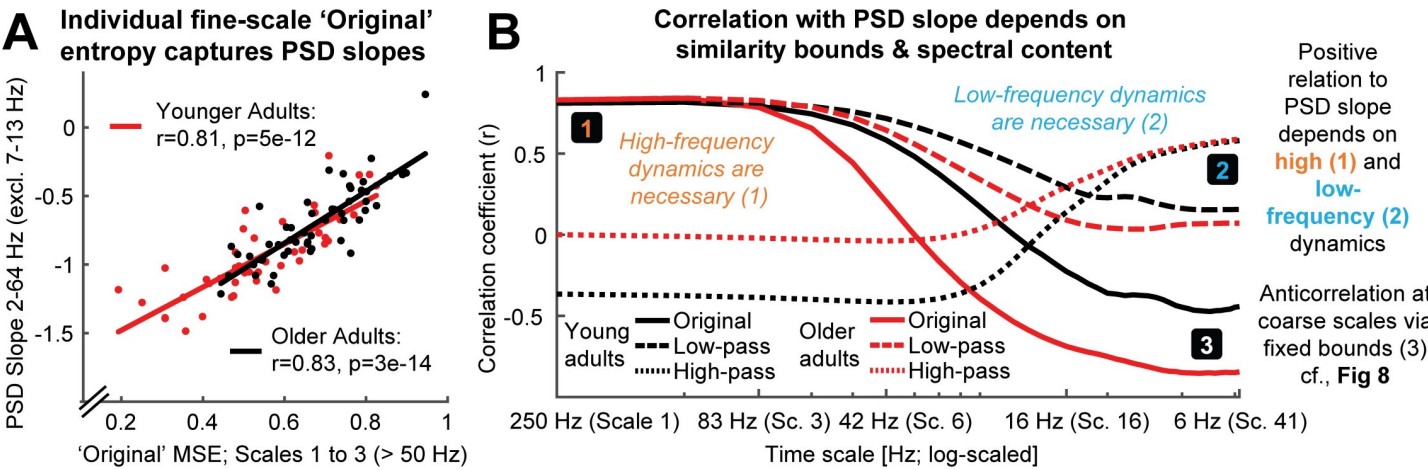

**Fig 9. The presence of low- and high-frequency content renders fine entropy slopes sensitive to PSD slopes.** A) Sample entropy at fine time scales represents the slope of power spectral density across age groups. The 7–13 Hz range was excluded prior to the PSD slope fit to exclude the rhythmic alpha peak (see Fig 8B). (B) The presence of both slow and fast dynamics is required for positive associations with PSD slopes to emerge. The direction and magnitude of correlations of scale-wise entropy with PSD slopes depends on the choice of global vs. rescaled similarity bounds, as well as the choice of filtering. Original entropy inverts from a positive correlation with PSD slope at fine scales to a negative association at coarse scales. Rescaling of the similarity bound abolishes the negative correlation of coarse-scale entropy with PSD slopes. S6 Fig presents scatter plots of these relationships. The x-axis indicates the upper frequency bounds for the low-pass version.

As expected, individual fine-scale entropy was strongly and positively related to PSD slopes (Fig 9A) in both younger and older adults. Notably, after high-pass filtering the signal, the positive relation of fine-scale entropy to PSD slopes disappeared in both age groups (Fig 9B, dotted lines), and turned negative in older adults (see S6 Fig for scatter plots), while age differences in fine-scale entropy disappeared (Fig 7D). Relations between entropy and PSD slopes–and age differences–re-emerged once low-frequency content was included in the entropy estimation (Fig 9B, dashed and dotted lines), indicating that the presence of slow fluctuations was necessary for PSD slope relations. To assess whether varying PSD slopes accounted for fine-scale age differences in 'Original' MSE, we computed partial correlations between the measures. No significant prediction of age group status by fine-scale entropy was observed when controlling for the high collinearity with PSD slopes (r = -.04, p = .69), whereas PSD slopes significantly predicted age group status when controlling for fine-scale entropy (r = .37, p = 2e-4).

Finally, spectral slopes were anticorrelated with coarse-scale entropy when global similarity bounds were used (Fig 9B, solid lines), but not when criteria were scale-wise re-estimated (Fig 9B, dashed and dotted lines). This again suggests a presence of the scale-wise bias noted in Issue 1 (i.e., scale-wise bound divergence); subjects with shallower slopes (more high frequency power) had increasingly liberally-biased thresholds at coarser scales, resulting in overly low entropy estimates.

In sum, age differences in fine-scale entropy were conditional on the presence of both low- and high-frequency dynamics and reflected differences in PSD slopes; while the pattern irregularity of fast dynamics *per se* was not modulated by age.

## Narrowband MSE indicates age differences in signal irregularity in alpha and beta band

The previous analyses highlighted how the spectral content of the signal can give rise to MSE time scale mismatches. However, our simulations also suggest a far more accurate mapping between entropy and power when scale-wise bandpass filters are used (Fig 4E). Concurrently, application of the band-pass implementation indicates a partial decoupling between entropy

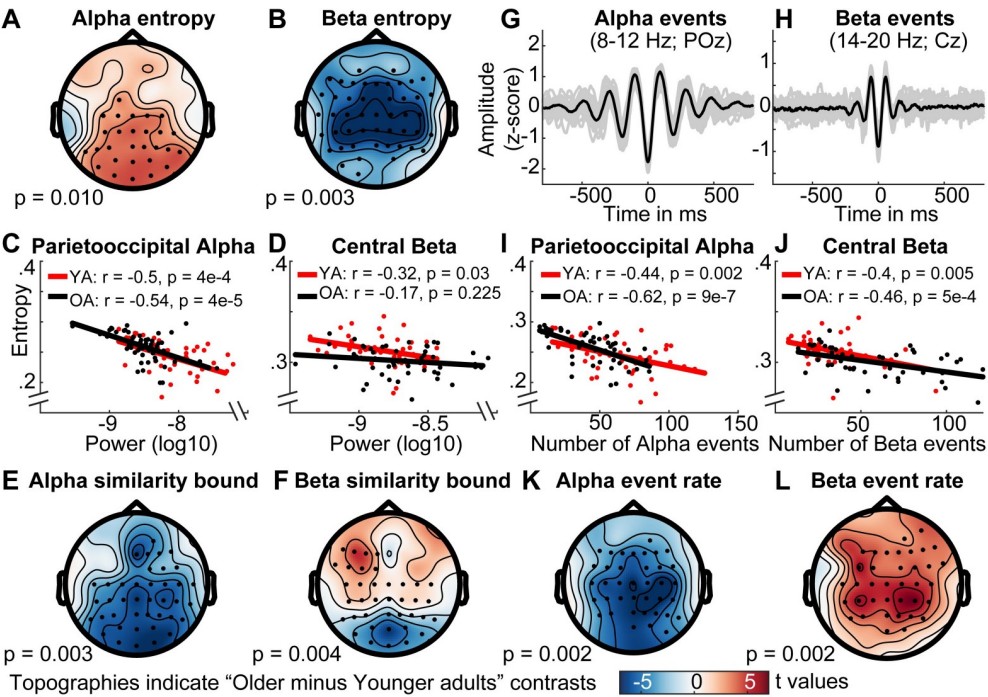

**Fig 10. Narrowband MSE reflects age differences in alpha- and beta-specific event (ir)regularity.** (A, B) Narrowband MSE indicates age differences in the pattern complexity at alpha (A) and beta (B) frequencies. (C, D) Alpha, but not beta power consistently correlates negatively with individual narrowband entropy within clusters of age differences. (E, F) Similarly, alpha but not beta similarity bounds show an inverted age effect with similar topography. (G, H) Single-trial rhythm detection highlights a more transient appearance of beta compared with alpha events. Data are collapsed across age groups. (I, J) The rate of stereotypical single-trial alpha and beta events is anticorrelated with individual narrowband entropy. (K, L) The rate of spectral events exhibits age differences that mirror those observed for entropy. Note that the same color range, plotted in the lower row, was plotted for all topographies.

and variance (as reflected in the *similarity bound*) age differences (Fig 7E). Specifically, older adults exhibited higher parieto-occipital entropy at alpha time scales (˜8–12 Hz) and lower central entropy at beta time scales (˜12–20 Hz) than younger adults (Fig 7; Fig 10A and 10B). Whereas alpha-band entropy was moderately and inversely correlated with alpha power (Fig 10C) and the age difference was inversely reflected in the similarity bound in a topographically similar fashion (Fig 10E), the same was not observed for entropy in the beta range for both age groups (Fig 10D and 10F). Promisingly, this indicates evidence for what many who employ MSE measures in cognitive neuroscience presume–that power and entropy *can* be decoupled, providing complementary signatures of neural dynamics.

This divergence of entropy and power in the beta band is particularly interesting as beta events have been observed to exhibit a more transient waveform shape [57, 58], while occupying a lower total duration during rest than alpha rhythms [42]. Indeed, it should be the rate of stereotypic spectral events that reduces pattern irregularity rather than the overall power within a frequency band. To better test this assumption in our data, we applied single-trial rhythm detection to extract the individual rate of alpha (8–12 Hz) and beta (14–20 Hz) events. As predicted, alpha events had a more sustained appearance compared with beta events as shown in Fig 10G and 10H (events were time-locked to the trough of individual events; see methods). Importantly, both alpha and beta event rate were inversely and moderately correlated with entropy estimates (Fig 10I and 10J) at matching time scales in the band-pass version. Correlations were also numerically higher than between power and entropy (Fig 10C and

10D), suggesting that entropy captured the non-stationary character of the rhythmic episodes that are not captured by sustained power estimates. The relationships remained stable after controlling for individual event rate and entropy in the age effect cluster of the other frequency band (partial correlations: alpha for younger adults: r = -.52, p = 2e-4; alpha for older adults: r = -.71, p = 8e-9; beta for younger adults r = -.49, p = 6e-4; beta for older adults: r = -.56, p = 2e-5), indicating separable associations between event rate and entropy between the two frequency bands. This is important, as our simulations suggest increased entropy estimates around narrow-band filtered rhythmicity (see Fig 4E). Furthermore, a permutation test indicated age differences in beta rate that were opposite in sign to the entropy age difference (see Fig 10L). In particular, older adults had a higher number of central beta events during the resting state compared with younger adults, thus rendering their beta-band dynamics more stereotypic. In sum, these results suggest that narrowband MSE estimates approximate the irregularity of non-stationary spectral events at matching time scales.

## Rhythmic alpha events transiently reduce broadband signal irregularity

Finally, the neurobiological relation between narrowband rhythms and broadband signal characteristics (spectral slopes in particular; Fig 9) is a substantive question of considerable interest [59–61]. Rhythmic alpha events have been theorized to phasically modulate cortical excitability, with higher amplitudes of alpha events thought to reflect an overall reduction in population activity due to reduced excitability [55]. Such activation levels in turn have been related to scale-free broadband characteristics in human electrophysiological data [38, 51–54], which strongly contribute to fine-scale entropy estimates (Fig 9; S7 Fig). It is thus conceivable that alpha rhythms transiently reduce broadband irregularity. In line with this notion, negative associations between alpha power and fine-scale entropy have been observed [40, 62]. However, sample entropy's joint sensitivity to broad- and narrowband dynamics ("Issue 2") (see Fig 4) makes it ambiguous whether such associations truly reflect shifts in broadband features. We confirm this ambiguity in simulations (Fig 11A; sample entropy calculated for 250 ms signals consisting of varying slope coefficients in the presence or absence of alpha rhythms), where we observe that increased rhythmic regularity during alpha events concurrently decreases sample entropy, even when no change has occurred in the aperiodic signal component (Fig 11A: red panels). Controlling the spectral signal content via band-stop filters (here: 8–15 Hz) removes such circular entropy decreases due to increased narrowband regularity in the alpha band, while accurately indicating entropy changes due to changes in spectral slopes (Fig 11: green panels).

We used fine-scale sample entropy's sensitivity to aperiodic slopes determined above (Fig 9; S7 Fig) to probe the relationship between broadband irregularity and rhythmic alpha events with high temporal precision in empirical data. To test transient modulations of irregularity during alpha rhythms, we leveraged the temporal on- and offsets of individual alpha segments (8–15 Hz; > 3 cycles) during eyes-open rest as uniquely identified by rhythm detection (see Fig 11B; see S8 Fig for exemplary traces). We created 250 ms segments surrounding the on- and offsets of alpha activity, followed by the calculation of sample entropy. To investigate potential differences as a function of magnitude, we median-split high- and low-amplitude alpha events. For both splits, we observed that sample entropy decreased upon alpha onset, whereas it recovered to high levels following alpha offset (Fig 11C1 and 11D1; red panels). However, due to the aforementioned circularity, the observation of transient entropy decreases during alpha periods offers little unambiguous insight beyond the successful identification of rhythmic event on- and offsets by the eBOSC algorithm. Importantly, transient entropy decreases during high-amplitude alpha events were also observed after removal of the alpha

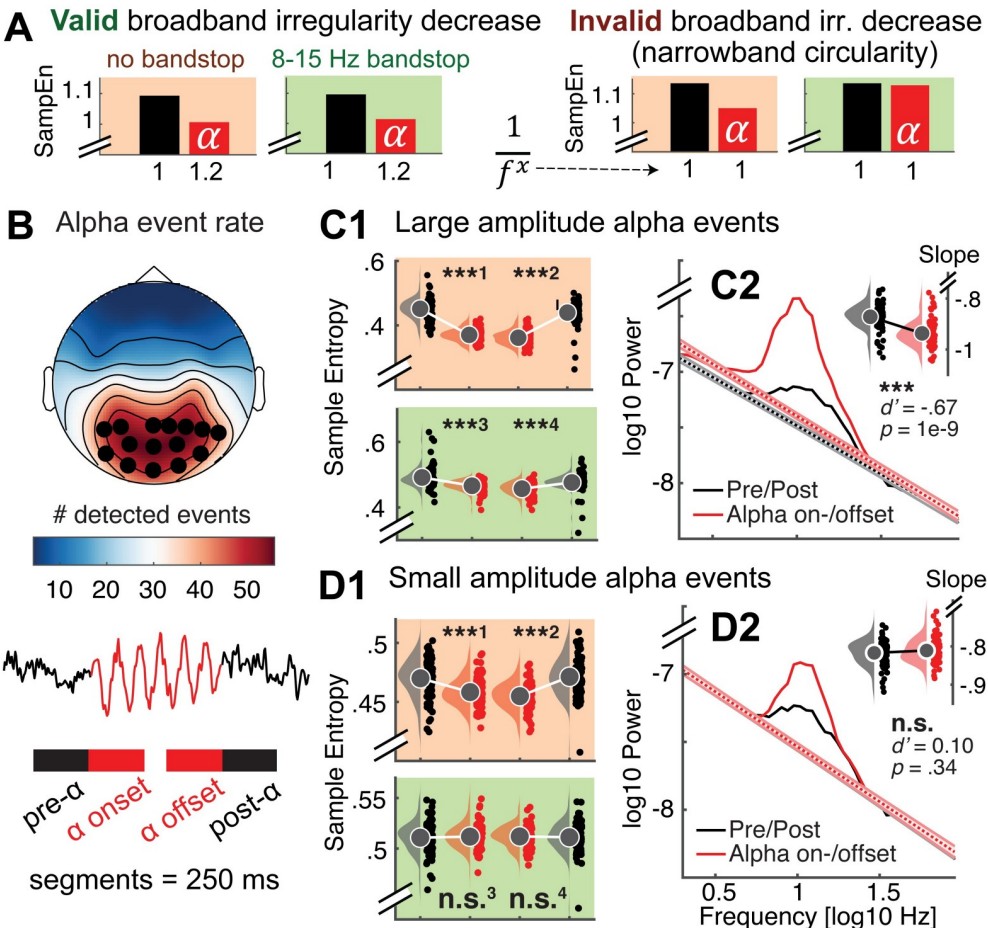

**Fig 11. Nonstationary alpha events transiently reduce broadband irregularity.** (**A**) Testing for transient broadband changes during alpha events requires control for narrowband circularity. We simulated 250 ms signals consisting of varying slope coefficients (plotted on the x-axis) in the presence or absence of alpha rhythms. Bars indicate first-scale entropy estimates (i.e., sample entropy; SampEn) for these signals, as well as bandstop-filtered versions. Left: Valid slope shallowing in the presence of alpha events was indicated both when alpha was included in estimates (red background), as well as when band-stop filters removed the influence of alpha regularity (green background). Right: Crucially, when no bandstop filters were applied, sample entropy decreased also in the absence of slope variations due to the added alpha regularity (red background). This effectively represents narrowband circularity in the analysis. In contrast, bandstop filters removed the influence of alpha regularity and permitted estimation of valid reductions in broadband irregularity (green background). (**B, C, D**) Empirical analysis of transient entropy decreases during alpha events. (B Alpha events were selected across channels with high amounts of detected events (black dots). Lower: Broadband entropy was calculated for 250 ms segments preceding and following the on- and offset of alpha events. (**C1**) During eyes open rest, nonstationary alpha events of high strength transiently reduce broadband irregularity, also after accounting for alpha circularity. Raincloud plots (RCPs) indicate the intervals schematically plotted in the bottom panel of B. For visualization, RCPs display estimates that are centered within-subject (condition-wise data minus individual across-condition average plus global average); statistics were calculated on uncentered estimates. ***[1]: $d' = -1.91$; $p \sim 0$. ***[2]: $d' = -1.61$; $p \sim 0$. ***[3]: $d' = -0.63$; p = 1e-8. ****[4] $d' = -0.54$; $p = 6e-7$ [d' = $(\bar{X}_{alpha} - \bar{X}_{pre/post})/STD(X_{alpha}-X_{pre/post})$]. (**C2**) Slope fits indicate a shallowing of slopes during alpha events. The inset bar plot indicates mean slopes estimates with within-subject standard errors. (**D1**) In contrast, irregularity decreases were indicated for low-amplitude alpha events only when circularity was not accounted for, but not after alpha was removed. This indicates that bandstop filtering successfully avoids circularity in empirical use cases. ***[1]: $d' = -0.52$; $p = 1e-6$. ***[2]: $d' = -0.75$; $p = 3e-11$. n.s.[3]: $d' = -0.05$; $p = 0.63$. n.s.[4] $d' = -0.04$; $p = 0.67$. (**D2**) No significant slope changes were observed during low-amplitude alpha events. Note that black dotted line is covered here. Error bars reflect within-subject standard errors.

band (Fig 11C1; green panel), indicating that narrowband amplitude increases in the alpha-band were not sufficient to explain the observed entropy differences. This provides evidence that spontaneous, large-amplitude alpha rhythms during eyes open rest transiently decrease broadband signal irregularity, supporting their suggested role in the modulation of cortical excitability. We did not observe an interaction between alpha status and age for any of the contrasts (all p > .05), suggesting that decreased irregularity during transient alpha events is a preserved characteristic of cortical alpha rhythms across the adult lifespan. To further investigate a broadband effect, we calculated spectral slopes (using an auto-sandwiching approach, see methods). This analysis revealed a transient steepening of slopes during alpha events, in line with a broadband shift towards decreased excitability (Fig 11C2). In contrast to high-amplitude events, entropy decreases were not indicated for low-amplitude events after accounting for circularity bias (Fig 11D1, green panel). Similarly, no shift in aperiodic slopes was observed (Fig 11D2). This suggests that the originally indicated entropy decreases during low-amplitude events do not represent broadband shifts. This analysis highlights sample entropy's potential to indicate fluctuations in signal irregularity with high temporal precision. Notably, the analysis reinforces the need for a targeted modulation of spectral content to avoid circular inferences, and reduce the ambiguity of results. Our findings suggest an alternative use case for dedicated bandpass filters that retains high sensitivity to broadband effects of interest. Specifically, the mechanistically informed use of band-stop filters here affords analyses into the modulators of signal irregularity and thereby can reveal non-trivial neurocomputational/-biological insights.

## Discussion

MSE aims to characterize the temporal irregularity of (neural) time series at multiple temporal scales. In the present study, we have highlighted two primary issues that may render the interpretation of time scales unintuitive in traditional applications: (Issue 1) biases from global similarity bounds, and; (Issue 2) the characterization of broadband, low-frequency dominated signals (see Fig 12A for a schematic summary). In the following, we discuss these effects and how they can impact traditional inferences regarding signal irregularity, in particular with regard to empirical age differences. Then, we discuss age effects in narrowband signal irregularity at interpretable temporal scales. Finally, we recommend procedures to improve scale-specific MSE inferences.

### Issue 1: Global similarity bounds bias coarse-scale entropy estimates

The ability to estimate entropy at coarser time scales provides the main motivation for a multi-scale implementation. Towards coarser scales, entropy is generally thought to represent the irregularity of increasingly slow dynamics. However, MSE's traditionally global similarity bounds systematically bias coarse scale entropy estimates. Given that scale-wise variance decreases across scales, the liberality of global similarity bounds increases, causing entropy to decrease despite no ostensible shift in pattern irregularity. This bias is independent of the values of the global similarity bound–which did not differ across groups here–but rather depends on the *removed* variance at the time scale of interest. This issue has led to puzzling results in past work. For example, several papers using 'original' MSE have shown that in white noise signals (which by definition should be equally irregular at all time scales due to their randomness), entropy unintuitively decreases towards coarser scales, whereas pink noise signals undergo less entropy reduction across initial scales due to the removal of less high-frequency content [29] (S7 Fig). Strikingly, such puzzling effects have been used to *validate* the most common implementation of MSE [e.g., 27, 32] rather than to indicate the presence of a

**Fig 12. Summary of the identified time-scale mismatches and recommendations for future studies.** (**A**) We highlight two scale-dependent mismatches that run counter to the intuition that entropy at fine scales primarily refers to fast dynamics, and vice-versa: (1) Coarse-scale entropy is biased towards reflecting high-frequency content when signals of decreasing variance are compared to a global, and increasingly inadequate, similarity bound. (2) Fine-scale entropy characterizes scale-free 1/f slopes when broadband signals include slow frequency content. Dashed colored arrows indicate the mismatched relations observed in the current study. (**B**) Beyond time-scale mismatches, brain signal entropy and variance/power can often be collinear, in part due to their shared description of linear signal characteristics, such as rhythmicity. To identify complementary and unique relations of pattern complexity compared to more established measures of variance, explicit statistical control is required for the latter. (**C**) We propose multiple strategies to safeguard future applications against the highlighted issues.

systematic bias in estimation. This appears motivated by the assumption that "changes of the variance due to the coarse-graining procedure are related to the temporal structure of the original time series, and should be accounted for by the entropy measure" [12]. We rather consider the similarity bound divergence a clear bias that invalidates the intuitive interpretation of time scales in MSE applications, and highlight that more intuitive broad-scale offsets are indicated when bound biases are removed (see S2 Text for elaboration on this issue).

Importantly, we highlight that this bias affects practical inferences. In the current resting-state EEG data, an age-related increase in high frequency power manifested unintuitively as a decrease in coarse-scale entropy via systematic group differences in the divergence of similarity bounds. Note that we presume that this age difference arises from a relative bias. As such, variations in high-frequency power suffice, even at low levels in 1/f scenarios, to systematically impact coarse-scale estimates and to specifically explain variance in a third variable of interest (e.g., age; see Fig 12B). Given that global similarity bounds remain prevalent in applications (see S1 Text), we hope that our practical example motivates the adoption of scale-varying parameters. Overall, we perceive little justification for the use of scale-invariant parameters in MSE estimation in future work. Indeed, as most previous work included biased, global bounds, reported coarse-scale effects may dominantly reflect false positives, while the sensitivity to true coarse-scale effects may have suffered, hence jointly increasing false negatives. Hence, results obtained with global bounds are ambiguous and hard to interpret. A critical task for future work (potentially including the re-analysis of existing data) will thus be to establish specific coarse-scale effects that provide empirical evidence for the practical utility of a multi-scale

entropy computation. Recent advances for the robust estimation of coarse-scale entropy from sparse neuroimaging data [34, 63, 64] may be required to better estimate coarse-scale effects in *in vivo* data.

## Issue 2: Fine-scale entropy relates to PSD slopes in the presence of slow frequency content

In parallel to the assumption of dominantly slow signal contributions to coarser scales, fine-scale entropy is often interpreted as a signature of "fast" temporal irregularity. However, it is typically estimated from broadband signals. As such, slow trends [35], neural rhythms at characteristic time scales [65] (Fig 4) and scale-free 'background' or 'noise' activity with a $\frac{1}{f^x}$ power-law form [38, 50, 53] (Fig 9; S7 Fig) jointly contribute to fine-scale entropy estimates. By linking fine-scale entropy to broadband PSD slopes, we replicated previous observations of increasing sample entropy with shallower slopes [14, 17, 29, 40, 46, 66] and shorter temporal autocorrelations [4, 27, 67]. However, we qualify this association by highlighting that the *joint* presence of slow and fast dynamics in the signal is necessary to produce such effects, hence verifying a broadband origin. At a mechanistic level, differences in spectral slopes and fine-scale entropy may jointly index variations in cortical excitability. Cortical neurons constantly receive a barrage of synaptic inputs. Variations in the excitatory and inhibitory summary statistics of these inputs robustly alter the conductance state of membrane potentials [for a review see 68], thereby producing variations in the irregularity of spike output and the appearance of global EEG signals [for a review see 69]. Whereas excitability is reduced during synchronized states characterized by strong low-frequency fluctuations, "desynchronized" cortical states feature enhanced sensitivity to external stimuli [70–72]. From a functional perspective, cortical information capacity, approximated via the entropy of cortical activity, may non-linearly vary alongside such excitation/inhibition (E/I) ratio, with highest information capacity afforded at intermediate levels of systemic excitability [10]. From a technical perspective, spectral (PSD) slopes have been proposed as a functional index of such an E/I ratio [49, 54, 73–75]. However, frequency-dependent filtering of current flow in the extracellular medium [76] or at the dendrite [77] may also contribute to the observed inter-individual differences in spectral slopes.

More generally, the association between broadband signal entropy and spectral slopes coheres with the notion that shallower slopes have a more 'noisy' or irregular appearance in the time domain. Thus, spectral slopes and temporal predictability are–at least in part–different perspectives on the same signal characteristic. Practically however, the correspondence between fine-scale entropy and 1/f slopes should nonetheless be tested, given that these scales are also sensitive to other signals characteristics, such as narrowband rhythmicity (Fig 4). Such necessity for narrowband control is highlighted by our analysis of transient fine-scale entropy changes during non-stationary alpha events (Fig 11). Only the removal of narrowband rhythmic regularity afforded non-circular insights. Specifically, we observed that broadband entropy transiently reduces following the onset and prior to the offset of parieto-occipital alpha rhythms, alongside a steepening of spectral slopes. This result is in line with alpha rhythms reflecting synchronized states with reduced cortical excitability [55, 59, 60, 78–81], but extends prior applications by characterizing non-stationary events at the single-trial level with high temporal precision, rather than temporal averages. Notably, our results contradict a prior observation that increased spontaneous alpha amplitudes at rest relate to a shallowing of low-frequency slopes, both in time and space [61]. Whether differences in frequency range, temporal specificity, or the stability of slope estimates contribute to this difference is an interesting question for future research that sample entropy may help to resolve. Notably, the fine-scale sensitivity of this effect highlights that single-scale broadband (sample) entropy–in the absence

of multiscale implementations–is *per se* sensitive to broadband effects of interest, benefitting applications with limited available data and time [e.g., closed-loop setups: 62].

## Spectral power and entropy: What's irregularity got to do with it?

For entropy to be a practical and non-redundant measure in cognitive neuroscience, both its convergent and discriminant validity to known signal characteristics should be established. Multiple features can influence the temporal irregularity of neural time series. These include traditional 'linear' PSD features, (e.g., temporal autocorrelation, rhythmicity, etc.) as well as 'non-linear' features (e.g., phase resets, cross-frequency coupling, etc.). It is therefore worth noting that associations between spectral power characteristics and entropy estimates are partly anticipated (Fig 12B). For example, as noted before, entropy should reduce with increased rhythmic irregularity, and increase with shallowing of PSD slopes (and hence, shortening of temporal autocorrelations). However, the use of MSE is often motivated by its partial sensitivity to non-linear properties of brain dynamics [27, 46] that cannot be captured by traditional PSD analyses [e.g., 82, 83, 84]. In extreme cases, the absence of linear contributions may be erroneously inferred from the use of variance-based similarity bounds. Contrary to such orthogonality assumptions, our analyses highlight that differences in spectral variance (as captured by the similarity bound, which is typically neglected as a measure of interest when estimating MSE) *can* account for a large proportion of reported MSE effects [see also appendix in 27]. As such, non-linear characteristics *per se* may often do little to drive MSE estimates (see also results from a surrogate analysis in S3 Text, S9 Fig). This is in line with dominant linear power contributions to non-linear measures [85]. Conversely, the specificity to valid and unique non-linear effects increases after methodologically accounting for linear contributions.

## Relevance of identified time scale mismatches to previous work

Although the highlighted issues broadly apply to applications in which MSE is a measure of interest (e.g., assessment of clinical outcomes [e.g., 22]; prediction of cognitive performance [e.g., 46]), our results are especially relevant for MSE differences across the lifespan. Previous applications indicated that older adults exhibit lower coarse-scale entropy and higher fine-scale entropy compared with younger adults [16, 18, 27, 86]. While we conceptually replicate these results with the standard MSE implementation, our analyses question the validity of previous interpretations. In particular, our results suggest that age-related increases in coarse-scale entropy do not reflect valid differences in the irregularity of slow dynamics, but rather reflect differential high frequency power biases [see also 19]. Moreover, our analyses ascribe age differences in fine-scale irregularity to a flattening of PSD slopes, as observed from child-to adulthood [46] and towards old age [16, 18, 40, 49]. Such shallowing of scale-free slopes suggests relative shifts from distributed to local processing, and coheres with the notion of increased "neural noise" due to increases in the local excitation/inhibition ratio [54].

Across development, altered time scales of neural computations (as indicated by broadband changes in autocorrelations) [87] may reflect changes in intra- and inter-cortical connectivity [88], arising from reductions in grey matter density [89, 90], the integrity of associative white matter tracts [91], and changes in local receptor distributions and neuromodulation [92–96]. Dynamic interactions between such morphological changes may jointly shape control over local excitability and 'neural noise' across the lifespan [97]. Two alternative functional consequences of developmental noise increases have been proposed. On the one hand, intermediate levels of noise may provide beneficial stochastic resonance effects [9, 98–100], in line with relations between higher entropy and behavioral benefits in child- and adulthood [46], as well as in older adults [86]. In contrast, overwhelming amounts of local noise can produce adverse

consequences [49, 101], supported by the observation that shallower slopes with advanced adult age relate to impaired working memory performance [49]. While further work including longitudinal assessments and behavioral probes will be necessary to disentangle the functional relevance of developmental changes, we argue that a principled separation of narrow- and broadband changes [102] will help to guide the search for neurobiological mechanisms driving entropy effects.

Taken together, our results suggest that entropy age differences dominantly arise from linear power differences, and appear at counterintuitive time scales. We confirmed the dominant contribution of age group differences in power characteristics using a surrogate analysis (see S3 Text, S9 Fig). Our surrogate analysis replicates a previous surrogate analysis that attributed age group differences mainly to linear auto-correlative properties [see appendix in 27, see also 85]. As we exclusively focused on univariate entropy, it remains an interesting question for future work whether our results are applicable to age-related decreases in 'distributed' entropy that capture the mutual information between distinct sensors [16].

## Cross-sectional age differences in narrowband MSE

Complementing traditional broadband applications, our use of narrowband MSE suggested age-related entropy increases in the posterior-occipital alpha band and decreases in central beta entropy that inversely tracked the regularity of alpha and beta events, respectively. Posterior-occipital decreases in alpha power and frequency with age are fundamental findings in many age-comparative studies [103]. While age-related increases in beta power are not observed as consistently [see e.g., 103 for a review], age-related increases in their prevalence have been observed during eyes open rest [104]. In addition, beta power increases over contralateral motor cortex during rest may reflect greater GABAergic inhibition in healthy aging [105]. While our results are not hemisphere-specific, they may similarly reflect increased inhibition in older adults, potentially reflected in an increased number of stereotypical beta events [58]. However, further work is required to establish the functional interpretation of narrowband age differences, as well as technical impacts of filter bandwidth, and individual center frequencies on narrowband results, especially given age differences in rhythmic peak frequencies [103]. Nevertheless, these results highlight that scale-specific narrowband filtering can provide novel, frequency-specific, insights into event/signal irregularity.

Notably, a narrowband approach may warrant different use cases than broadband entropy. In particular, the sensitivity to multi-scale information, such as cross-frequency interactions and waveform shape, is a defining characteristic of (and motivation for using) entropy as opposed to spectral analysis. However, this sensitivity trades off with specificity when a narrowband approach is chosen, which by definition enforces a more rhythmic appearance than the raw signal may convey [106]. Nonetheless, frequency-specific phenomena such as variations in the amplitude or the presence of rhythmic events are complementary signatures of irregularity in their own right. For example, long-range temporal correlations (LRTCs) of narrowband amplitudes provide an alternative window on the irregularity of temporal dynamics [107–109]. As such, targeted filter applications–either chosen *a priori* or as a follow-up to broadband entropy effects–may prove useful to delineate spectrally specific effects at directly interpretable neural time scales. Hence, we do not regard narrowband MSE as a replacement for the traditional low-pass implementation of MSE, but rather as a parallel tool for the exploration and probing of broadband effects. Moreover, sensitivity to broad-scale phenomena remains high when band-stop filters are used (e.g., Fig 11), highlighting the general feasibility of applying narrowband filters to derive broadband insights beyond the band-stop range.

### Recommendations for future applications

The issues raised here suggest that additional steps need to be taken to achieve valid scale-wise estimates of MSE, and to support the perceived complementary nature of MSE relative to more typical measures (such as spectral power, etc.). We are optimistic that the following recommendations Fig 12C, which have already been partially proposed [33–35, 63, 110], improve the utility of MSE as a principled tool for the estimation of complex brain dynamics.

a. We see little motivation for the use of global similarity bounds as they introduce challenges rather than benefits. We therefore recommend the MSE field to abandon *global* similarity bounds in favor of scale-specific bounds. We hope that our showcase of their detrimental consequences contributes to the wide-scale adoption of 'refined' approaches [e.g., 33, 34, 110], which we consider the minimum requirement for novel neurocomputational insights.

b. We recommend spectral filters to validate the scale-specificity and/or broadband nature of effects. For example, if effects are observed at fine temporal scales with a low-pass filter, additional high-pass filters may inform about the spectral extent of the effect. For entropy estimates of slow dynamics, traditional low-pass filter settings already apply this principle by becoming increasingly specific to slow fluctuations (if scale-dependent normalization is used)–but crucially, specify to high-frequency content is never attained. This proposal represents a general extension of proposed solutions based on high-pass filtering to remove slow trends [35], or based on incorporating slow temporal correlations into parametric models for the MSE estimation [34, 63].

c. We regard statistical control as necessary to establish entropy effects that are not capturable by traditional linear indices (such as PSD characteristics). While some studies have shown joint effects of interest in MSE and (band-limited) spectral power [15, 16, 18, 19, 111–117], others identified unique MSE effects [22, 118–120]. However, the (mis)match between time-scales and frequencies may not always be readily apparent, at least in part due to the various issues raised here. As shown here, controls should include both narrowband ('rhythmic') power and the arrhythmic signal background. As the scale-wise *similarity bound* is used for normalization, it should at the very least be controlled for. The choice of features may further be aided by comparing effect topographies of spectral power and entropy, as done in the present study. An important point to note is the relevance of statistical controls for relations to third variables (see Fig 12B). While some studies highlight scale-dependent associations of entropy with power, a large amount of shared variance (e.g., of coarse-scale entropy with slow frequency power) does not guarantee that a smaller portion of residual variance (e.g., shared with normalization biases) systematically does or does not relate to other effects of interest. This is equally relevant for identifying unique non-linear contributions. For example, while we observed moderate associations between band-specific rhythm events and entropy here, this non-redundant association nevertheless leaves room for the two measures to diverge in relation to third variables. This is in line with prior work [27, 121] showing that despite a dominant influence of linear characteristics on entropy estimates, non-linear contributions can uniquely explain a (smaller) portion of entropy variance.

d. Finally, a principled way to dissociate non-linear signal characteristics from linear signal variance is to use phase-shuffled surrogate data [5, 122–125]. Phase randomization (see S3 Text, S9 Fig) effectively alters original time series patterns while preserving linear PSD characteristics and "is unavoidable if conclusions are to be drawn about the existence of nonlinear dynamics in the underlying system" [5]. While such surrogate approaches have been

utilized in select entropy applications [4, e.g., appendix of 27] to highlight entropy's non-linear sensitivity [e.g., 30, 32, 46], it has not become common practice in application. Given that MSE is sensitive to many linear characteristics, some of which are shown in the present work, we consider surrogate analyses as an optimal approach to verify the contribution of non-linear signal characteristics.

In combination, such controls may go a long way toward establishing unique, complementary, and valid contributions of MSE in future work.

## Conclusions

Many inferences regarding multiscale entropy in cognitive/clinical neuroscience rely on the assumption that estimates uniquely relate to pattern irregularity at specific temporal scales. Here we show that both assumptions may be invalid depending on the consideration of signal normalization and spectral content. Using simulations and empirical examples, we showed how spectral power differences can introduce entropy effects that are inversely mapped in time scale (i.e., differences in the high frequency power may be reflected in coarse entropy and vice versa; see Fig 12A). As these results suggest fundamental challenges to traditional MSE analysis procedures and inferences, we highlight the need to test for unique entropy effects (Fig 12B) and recommend best practices and sanity checks (Fig 12C) to increase confidence in the complementary value of pattern irregularity for cognitive/clinical neuroscience. While the warranted claim has been made that "it would be unreasonable simply to reduce sample entropy to autocorrelation, spectral power, non-stationarity or any of their combinations" [4], this should not mean that we cannot test whether one or more of these contributors may sufficiently explain MSE effects of interest. We thus propose that MSE effects may be taken as a starting point to explore the linear and nonlinear features of brain signals [e.g., 126]. We believe that empirical identification of the unique predictive utility of MSE will advance the quest for reliable mechanistic indicators of flexible brain function across the lifespan, and in relation to cognition, health, and disease.

## Methods

### Simulations of relations between rhythmic frequency, amplitude, and MSE

To assess the influence of rhythmicity on entropy estimates, we simulated varying amplitudes (0 to 7 arbitrary units in steps of 0.5) of 10 Hz (alpha) rhythms on a fixed 1/f background. This range varies from the absence to the clear presence of rhythmicity (see S3 Fig for an example). The background consisted of $\frac{1}{f^x}$-filtered Gaussian white noise (mean = 0; std = 1) with x = 1 that was generated using the function f_alpha_gaussian [127]. The background was additionally band-pass filtered between .5 and 70 Hz using 4$^{th}$ order Butterworth filters. Eight second segments (250 Hz sampling rate) were simulated for 100 artificial, background-varying trials, and phase-locked 10 Hz sinusoids were superimposed. To analyze the reflection of rhythmic frequency on time scales and to replicate a previously observed linear frequency-to-timescale mapping between the spectral and entropy domains [4, 22, 44], we repeated our simulations with sinusoids of different frequencies (5 Hz, 10 Hz, 20 Hz, 40 Hz, 80 Hz), that covered the entire eight second-long segments. For a specified amplitude level, the magnitude of frequency-specific power increases (or narrowband signal-to-noise ratio) increased alongside simulated frequencies due to the decreasing frequency power of pink noise, while the ratio of rhythmic-to-global signal variance (or global signal-to-noise ratio (SNR)) remained constant across simulated frequencies. We used the following definition: $\text{SNR}_{\text{global}} = \left(\frac{RMS_{signal}}{RMS_{noise}}\right)^2$, where

$RMS_{noise}$ is the root mean square of the pink noise time series and $RMS_{signal}$ characterizes the pink noise signal with added rhythmicity.

## Resting state data and preprocessing

To investigate the influence of similarity bounds and filter ranges in empirical data, we used resting-state EEG data collected in the context of a larger assessment prior to task performance and immediately following electrode preparation. Following exclusion of three subjects due to recording errors, the final sample contained 47 younger (mean age = 25.8 years, SD = 4.6, range 18 to 35 years; 25 women) and 52 older adults (mean age = 68.7 years, SD = 4.2, range 59 to 78 years; 28 women) recruited from the participant database of the Max Planck Institute for Human Development, Berlin, Germany (MPIB). Participants were right-handed, as assessed with a modified version of the Edinburgh Handedness Inventory [128], and had normal or corrected-to-normal vision. Participants reported to be in good health with no known history of neurological or psychiatric incidences, and were paid for their participation (10 € per hour). All older adults had Mini Mental State Examination (MMSE) [129, 130] scores above 25. All participants gave written informed consent according to the institutional guidelines of the Deutsche Gesellschaft für Psychologie (DGPS) ethics board, which approved the study.

Participants were seated at a distance of 80 cm in front of a 60 Hz LCD monitor in an acoustically and electrically shielded chamber. Following electrode placement, participants were instructed to rest for 3 minutes with their eyes open and closed, respectively. During the eyes open interval, subjects were instructed to fixate on a centrally presented fixation cross. An auditory beep indicated to the subjects when to close their eyes. Only data from the eyes open resting state were analyzed here. EEG was continuously recorded from 64 active (Ag/AgCl) electrodes using BrainAmp amplifiers (Brain Products GmbH, Gilching, Germany). Sixty scalp electrodes were arranged within an elastic cap (EASYCAP GmbH, Herrsching, Germany) according to the 10% system [131], with the ground placed at AFz. To monitor eye movements, two electrodes were placed on the outer canthi (horizontal EOG) and one electrode below the left eye (vertical EOG). During recording, all electrodes were referenced to the right mastoid electrode, while the left mastoid electrode was recorded as an additional channel. Online, signals were digitized at a sampling rate of 1 kHz.

Preprocessing and analysis of EEG data were conducted with the FieldTrip toolbox [132] and using custom-written MATLAB (The MathWorks Inc., Natick, MA, USA) code. Offline, EEG data were filtered using a 4th order Butterworth filter with a pass-band of 0.2 to 125 Hz. Subsequently, data were downsampled to 500 Hz and all channels were re-referenced to mathematically averaged mastoids. Blink, movement and heart-beat artifacts were identified using Independent Component Analysis [ICA; 133] and removed from the signal. Artifact-contaminated channels (determined across epochs) were automatically detected using (a) the FASTER algorithm [134], and by (b) detecting outliers exceeding three standard deviations of the kurtosis of the distribution of power values in each epoch within low (0.2–2 Hz) or high (30–100 Hz) frequency bands, respectively. Rejected channels were interpolated using spherical splines [135]. Subsequently, noisy epochs were likewise excluded based on FASTER and on recursive outlier detection. Finally, recordings were segmented to participant cues to open their eyes, and were epoched into non-overlapping 3 second pseudo-trials. To enhance spatial specificity, scalp current density estimates were derived via 4th order spherical splines [135] using a standard 10–05 channel layout (conductivity: 0.33 S/m; regularization: 1^-05; 14th degree polynomials).

## Calculation of (modified) multi-scale sample entropy (mMSE)

MSE characterizes signal irregularity at multiple time scales by estimating sample entropy (SampEn) at each time scale of interest. A schematic of the estimation pipeline is shown in S1 Fig. The mMSE code is provided at https://github.com/LNDG/mMSE. A tutorial for computing mMSE has been published on the FieldTrip website (http://www.fieldtriptoolbox.org/example/entropy_analysis/).

**Sample entropy estimation procedure.** The estimation of SampEn involves counting how often patterns of $m$ successive data points reoccur in time ($p^m$) and assessing how many of those patterns remain similar when the next sample $m+1$ is added to the sequence ($p^{m+1}$). Given that amplitude values are rarely exactly equal in physiological time series, a *similarity bound* defines which individual data points are considered similar. This step discretizes the data and allows to compare data patterns rather than exact data values. The similarity bound is defined as a proportion $r$ of the time series standard deviation (*SD*; i.e., square root of signal variance) to normalize the estimation of sample entropy for total signal variation. That is, for any data point $k$, all data points within $k \pm r \times SD$ are by definition equal to $k$, which forms the basis for assessing sequence patterns. SampEn is finally given as the natural log of $p^m(r)/p^{m+1}(r)$. Consequently, high SampEn values indicate low temporal regularity as many patterns of length $m$ are not repeated at length $m+1$. In our applications, $m$ was set to 2 and r was set to .5, in line with prior recommendations [13] and EEG applications [27, 46, 136].

**Multi-scale signal derivation procedure.** To extend sample entropy to multiple time scales, MSE 'coarse-grains' the original time series for multiple scale factors $\tau$ (here 1 to 42, where 1 refers to the original signal). The 'Original' MSE method [11, 12] averages time points within non-overlapping time bins (i.e., 'point averaging'). Such point averaging is equivalent to a low-pass finite-impulse response (FIR) filter, which can introduce aliasing however [33, 137] and constrains the specificity towards increasingly slow signals, while not allowing specificity to fast dynamics or any particular frequency range of interest. To implement control over the scale-wise filter direction and to reduce aliasing, we applied either low- [31, 33, 137], high-, or band-pass filters at each scale factor. The low-pass cut-off was defined as LP = $\frac{1}{scale} * nyquist$ and was implemented using a 6th order Butterworth filter, with nyquist defined as half the sampling rate of the signal. Similarly, the high-pass cut-off was defined as HP = $\frac{1}{scale+1} * nyquist$, implemented via 6th order Butterworth filters. Note that these cut-offs describe the upper and lower frequency bounds at each time scale, respectively. Finally, band-pass filters were applied to obtain narrowband estimates by sequentially applying Chebyshev Type I low- and high-pass filters (4th order with passband ripple of 1dB; chosen to achieve a fast filter roll-off), thus ensuring that each scale captured frequency-specific information. The passband was defined as BP = $LP$ +- 0.05*$LP$. To avoid pronounced passband ripple for broad passbands, 10th order Butterworth filters replaced the Chebyshev filters at scales where the passband was larger than 0.5*Nyquist. At scale 1, only a high-pass 10th order Butterworth filter was applied as the sampling rate of the signal set the upper (Nyquist) frequency bound. These settings were chosen to optimize the pass-through of signals within the pass-band and the attenuation of signals outside the pass-band. Two-pass filtering using MATLAB's filtfilt function was applied to achieve zero-phase delay. S4 Fig shows the spectral attenuation properties [138] of the filters. To avoid edge artefacts, input signals were symmetrically mean-padded with half the pseudo-trial duration (i.e., 1500 ms). After filtering, we implemented a point-skipping procedure to down-sample scale-wise signals (see S1 Fig). Since point-skipping allows for increasing starting point permutations $k$ for increasing scale factors $\tau$, we counted patterns separately for each starting point $k$, summed the counts of pattern matches and non-matches across them, and computed sample entropy based on the summed counts as described above:

$MSE(\mathbf{x}, \tau, \mathbf{m}, \mathbf{r}) = ln\left(\frac{\sum_{k=1}^{\tau} p^m}{\sum_{k=1}^{\tau} p^{m+1}}\right)$. This implementation is equivalent to "refined composite MSE" [110] and can improve the stability of entropy results for short or noisy signals [31, 110]. Note that no point skipping was performed in the 'high-pass' implementation to avoid low-pass filtering. As a result, the signals at increasing scale factors remained at the original sampling rate. To alleviate computational cost, scale factors were sampled in step sizes of 3 for empirical data (only for the 'high-pass' implementation) and later spline-interpolated. An adapted version of MSE calculations was used for all settings [64], in which scale-wise entropy was estimated across discontinuous data segments. The estimation of scale-wise entropy across trials allows for reliable estimation of coarse-scale entropy without requiring long, continuous signals, while quickly converging with estimates from continuous segments [64].

**Multi-scale calculation of similarity bounds.** Following scale-specific filtering, all implementations re-calculated sample entropy for the scale-specific signal. Crucially, in 'Original' applications [11, 12], the *similarity bound* is calculated only once from the original broadband signal. As a result of filtering, the scale-wise signal SD decreases relative to the global, scale-invariant similarity bound [29]. To overcome this limitation, we recomputed the similarity bound for each scale factor, thereby normalizing MSE with respect to changes in overall time series variation at each scale (.5 x SD of scale-wise signal).

**Scale factor notation.** As the interpretation of estimates at each scale is bound to the scale-wise spectral content, our Figures indicate spectral bounds of the scale-wise signals alongside the scale factor as follows: for the low- and band-pass implementation, we indicate the low-pass frequency as calculated above as the highest resolvable (i.e., Nyquist) frequency in the scale-specific signal. Likewise, for the high-pass implementation, we indicate the high-pass limit as the lowest resolvable frequency in the scale-specific signal. In the main text, we refer to higher scale factors as 'coarser' scales and lower scale factors as 'finer' scales, in line with the common use in the literature. Note that the sampling rate of the simulated data was 250 Hz, whereas the empirical data had a sampling rate of 500 Hz.

## Calculation of power spectral density (PSD)

Power spectral density estimates were computed by means of a Fast Fourier Transform (FFT) over 3 second pseudo-trials for 41 logarithmically spaced frequencies between 2 and 64 Hz (employing a Hanning-taper; segments zero-padded to 10 seconds) and subsequently averaged. Spectral power was $log_{10}$-transformed to render power values more normally distributed across subjects. Power spectral density (PSD) slopes were derived by linearly regressing power values on $log_{10}$-transformed frequencies (i.e., log-log fit). The spectral range from 7–13 Hz was excluded from the background fit to exclude a bias by the narrowband alpha peak [40, 49].

## Detection of single-trial spectral events

Spectral power, even in the narrowband case, is unspecific to the occurrence of systematic rhythmic events as it also characterizes periods of absent rhythmicity [e.g., 139]. Specifically detecting rhythmic episodes in the ongoing signal alleviates this problem, as periods of absent rhythmicity are excluded. To investigate the potential relation between the occurrence of stereotypic spectral events and narrowband entropy, we detected single-trial spectral events using the extended BOSC method [42, 140, 141] and probed their relation to individual entropy estimates. In short, this method identifies stereotypic 'rhythmic' events at the single-trial level, with the assumption that such events have significantly higher power than the 1/f background and occur for a minimum number of cycles at a particular frequency. This effectively dissociates narrowband spectral peaks from the arrhythmic background spectrum. Here, we used a

one cycle threshold during detection, while defining the power threshold as the 95[th] percentile above the individual background power. A 5-cycle wavelet was used to provide the time-frequency transformations for 49 logarithmically-spaced center frequencies between 1 and 64 Hz. Rhythmic episodes were detected as described in [42]. Following the detection of spectral events, the rate of spectral episodes longer than 3 cycles was computed by counting the number of episodes with a mean frequency that fell in a moving window of 3 adjacent center frequencies. This produced a channel-by-frequency representation of spectral event rates, which were the basis for subsequent significance testing. Event rates and statistical results were averaged within frequency bins from 8–12 Hz (alpha) and 14–20 Hz (beta) to assess relations to narrowband entropy and for the visualization of topographies. To visualize the stereotypic depiction of single-trial alpha and beta events, the original time series were time-locked to the trough of individual spectral episodes and averaged across events [c.f., 57]. More specifically, the trough was chosen to be the local minimum during the spectral episode that was closest to the maximum power of the wavelet-transformed signal. To better estimate the local minimum, the signal was low-pass filtered at 25 Hz for alpha and bandpass-filtered between 10 and 25 Hz for beta using a 6[th] order Butterworth filter. A post-hoc duration threshold of one cycle was used for the visualization of beta events, whereas a three-cycle criterion was used to visualize alpha events. Alpha and beta events were visualized at channels POz and Cz, respectively.

## Examination of transient irregularity shifts during alpha events

The relation of narrowband alpha events to broadband irregularity represents an empirical question of interest (see Introduction). We examined the relation between these signatures, while controlling for the circular, intrinsic relation between alpha-based regularity and entropy. To highlight the issue of circularity, we first simulated expected links between the two signals by creating 250 ms of data, consisting of (a) aperiodic slopes of $\frac{1}{f^1}$, (b) aperiodic slopes of $\frac{1}{f^{1.2}}$, as well as equivalent versions with superimposed alpha rhythms of unit amplitude (c, d). We probed the practical potential of a 8–15 Hz band-stop filter (6[th] order Butterworth) to remove the influence of alpha on broadband entropy. Entropy was calculated for the first MSE scale, reflecting broadband sample entropy. Next, in empirical data, we leveraged the temporal on- and offsets of individual alpha segments (8–15 Hz; > 3 cycles) as identified via rhythm detection and segmented the original data to include 250 ms preceding and following event on- and offsets (see S8 Fig for empirical examples). For each subject, all events across posterior-occipital channels at which event number was highest (see Fig 11B) were included in this analysis. At each channel we performed a median split of events according to their amplitude (high/low). We created versions with and without application of 8–15 Hz bandstop filters (S8 Fig), followed by the calculation of sample entropy. We assessed the impact of transient alpha events on irregularity via paired t-tests between alpha on vs. off contrasts, both at event on- and the offset, and individually for low and high amplitude events. As post-hoc tests, we assessed potential interactions between alpha presence and age via linear mixed effect models (random subject intercept). To probe the presence of a broadband effect, we assessed the spectral slopes for the same segments. To improve spectral resolution, we"auto-sandwiched" each 250 ms segment by appending it in x- & y-inverted forms at the original segment's on- and offset. This effectively increased segment duration to 750 ms, while retaining autocorrelative properties. We then calculated an FFT of each segment (2–90 Hz; 45 $2^x$ steps; Hanning taper; 4 Hz smoothing box; zero-padded to 10 s). Linear slopes were fit in log-log space, after excluding the 5–20 Hz range to remove the influence of the rhythmic alpha peak. Individual entropy estimates were averaged across alpha on- and offsets to remove measurement noise, and were statistically compared between alpha on & off periods via paired t-tests.

## Statistical analyses

Spectral power and entropy were compared across age groups within condition by means of independent samples t-tests; cluster-based permutation tests [142] were performed to control for multiple comparisons. Initially, a clustering algorithm formed clusters based on significant t-tests of individual data points (p < .05, two-sided; cluster entry threshold) with the spatial constraint of a cluster covering a minimum of three neighboring channels. Then, the significance of the observed cluster-level statistic, based on the summed t-values within the cluster, was assessed by comparison to the distribution of all permutation-based cluster-level statistics. The final cluster p-value that we report in figures was assessed as the proportion of 1000 Monte Carlo iterations in which the cluster-level statistic was exceeded. Cluster significance was indicated by p-values below .025 (two-sided cluster significance threshold). Effect sizes for MSE age differences with different filter settings were computed on the basis of the cluster results in the 'Original' version. This was also the case for analyses of partial correlations. Raw MSE values were extracted from channels with indicated age differences at the initial three scales 1–3 (>65 Hz) for fine MSE and scales 39–41 (<6.5 Hz) for coarse MSE. $R^2$ was calculated based on the t-values of an unpaired t-test: $R^2 = \frac{t^2}{t^2+df}$ [143]. The measure describes the variance in the age difference explained by the measure of interest, with the square root being identical to Pearson's correlation coefficient between continuous individual values and binary age group. Effect sizes were compared using the r-to-z-transform and a successive comparison of the z-value difference against zero: $Z_{Diff} = \frac{z1-z2}{sqrt\left(\frac{1}{N1-3}+\frac{1}{N2-3}\right)}$ [144]. Unmasked t-values are presented in support of the assessment of raw statistics in our data [145].

## Supporting information

**S1 Text. Systematic literature search assessing the prevalence of global similarity bounds.** (PDF)

**S2 Text. Simulation of MSE's sensitivity to PSD slope variation.** (PDF)

**S3 Text. Surrogate analysis of age effects.** (PDF)

**S1 Fig. Overview of modified (mMSE) adaptations.** First, mMSE uses data aggregation across (here: pseudo-) trials to allow the estimation of coarse scales also from sparse neuroimaging data [64]. These aggregated signals are then filtered at each scale prior to sample entropy calculation. The 'Original' implementation uses 'point averaging' for different scale factors, which is equivalent to a FIR low-pass filter. In adapted applications, we used a two-step implementation, which we refer to as 'filt-skip', which first applies a scale-wise low-, high- or band-pass filter, and then performs point skipping to down-sample the resulting signals. Finally, the sample entropy of these signals is similarly assessed using the sample entropy algorithm, which results in multiscale entropy estimates. Figure adapted with permission from [121]. (TIF)

**S2 Fig. Liberal similarity bounds reduce sample entropy in simulations.** (**A**) The plot shows the sample entropy of simulated white noise signals with constant signal standard deviation (SD) of 1, but varying similarity bounds. We denote this as a function of a scaling factor (SF) to highlight that such variation may arise from either variation in r, SD or both. Note that the r parameter is usually fixed and the SD matches the signal SD (gray line), thus normalizing total signal variance. However, when the similarity bound systematically increases relative to the

signal SD, entropy estimates progressively decrease (black line). (**B**) A similar scenario applies when fixed and large bounds are applied to signals of decreasing variance, as is the case across MSE time scales due to scale-wise filtering (Fig 2). Whereas no bias is observed when scale-wise signal SD is used for the calculation of similarity bounds (grey line), entropy estimates systematically decrease when the SD of the original signal are used (black line). Hence, the mismatched similarity bounds introduced entropy decreases although no changes to the structure of the (here white noise) signals were introduced.
(TIF)

**S3 Fig. Examples of simulated rhythmicity projected into pink noise.** (**A**) Top-down view of time-series from an exemplary simulated trial for a pure 1/f signal pink noise signal and at different magnitudes of added alpha rhythmicity. (**B**) Exemplary time series in 2D view. The red time series indicates an example time series for the level of rhythmicity shown in Fig 5. (**C**) Simulated SNR as a function of amplitude level. The dots indicate SNR for the levels depicted in panel B.
(TIF)

**S4 Fig. Filter magnitude responses.** (**A**) Filter magnitude responses at 10 Hz. Note that magnitude responses have been squared due to two-pass filtering to achieve zero-phase offsets. (**B**) Filter magnitudes of Bandpass filters (3$^{rd}$ order type I Chebyshev filter with 1dB passband ripple) at different time scales (red-to-orange indicating fine-to-coarse time scales). Note that only a high-pass filter (6$^{th}$ order Butterworth filter) is applied at the first scale.
(TIF)

**S5 Fig. T-values for age group differences in spectral power (OA > YA).** Statistical significance ($p < .05$) was assessed by means of cluster-based permutation tests and is indicated via opacity.
(TIF)

**S6 Fig. Methods- and scale-dependent associations between sample entropy and PSD slopes.** 'Original' settings indicate a strong positive association at fine scales (A1) that turns negative at coarse scales (A2), likely due to coarse-scale biases by the scale-invariant similarity criterion. In line with this notion, scale-wise adaptation of thresholds retains the fine-scale effect (B1), while abolishing the coarse-scale inversion (B2). Crucially, the entropy of exclusively high-frequency signals does not positively relate to PSD slopes (C1), whereas the association reemerges once slow fluctuations are added into the signal (C2).
(TIF)

**S7 Fig. Results of different simulated spectral slope coefficients for the different filter implementations.** (**A**) Using traditional implementations, 1/f variation introduces scale-dependent crossover effects, including scale-dependent entropy decreases for the signals approaching white noise. (**B, C, D**) In contrast, control for scale-wise variance indicates broad scale entropy offsets without crossovers. (**E**) Bandpass entropy is not modulated by broadband effects, as expected by the absence of multi-scale information at local scales.
(TIF)

**S8 Fig. Signal traces around indicated large alpha event on- and offsets.** (**A**) Thirty randomly selected traces across subjects for alpha on- (A1) and offsets (A2). The grey background indicates the 250 ms pre- and post-alpha windows used for the calculation of sample entropy (see Fig 11). The red background highlights segments following indicated alpha onsets, and preceding alpha offsets, that were used to assess irregularity during transient alpha events. Note that 250 ms segments may overlap in the case of short rhythmicity of around 3 cycles. (**B**)

All events around on- and offsets. Data were sorted by the instantaneous phase at +100 ms after indicated alpha onset (B1) and -100 ms prior to indicated alpha offset (B2). Instantaneous phase was calculated from a Hilbert transform applied to 8–15 Hz bandpass filtered signals. (**C**) Same as in B, but plotted for signals after 8–15 Hz bandstop filter application. All displayed traces were z-scored for presentation purposes.
(TIF)

**S9 Fig. Results of surrogate analysis, testing for non-linear contributions to MSE age effects.** (**A**) Examples of original and surrogate data for a random 3 s segment from an occipital channel with strong alpha rhythms. Phase randomization alters higher-order (non-linear) frequency interactions while preserving the linear power characteristics of the original data. If non-linear contributions are necessary for MSE age effects, no age effects should be indicated for entropy estimates of surrogate data (**B**) Results for "Original" MSE analysis on phase-shuffled data indicate similar effects as observed for original data (Fig 7A), suggesting that linear characteristics were sufficient for the observed age effects. (**C**) Results for low-pass MSE analysis on phase-shuffled data indicate similar effects as observed for original data (Fig 7C), suggesting that linear characteristics were sufficient for the observed age effects. (**D, E**) In addition to assessing the necessity of non-linear contributions, we further assessed whether age differences would be indicated for non-linear contributions, after accounting for linear power characteristics. The ratio of MSE estimates for original vs. surrogate data indicates unique non-linear contributions for either age group. The obtained results were remarkably similar for both original (D) and low-pass implementations (E), indicating the successful elimination of power-based biases. However, no statistically significant age differences were indicated, suggesting that non-linear contributions are at most minor, and may require higher statistical power for their assessment.
(TIF)

## Acknowledgments

We thank our research assistants and participants for their contributions to the present work.

## Author Contributions

**Conceptualization:** Julian Q. Kosciessa, Niels A. Kloosterman, Douglas D. Garrett.

**Data curation:** Julian Q. Kosciessa, Niels A. Kloosterman.

**Formal analysis:** Julian Q. Kosciessa.

**Funding acquisition:** Douglas D. Garrett.

**Investigation:** Julian Q. Kosciessa, Douglas D. Garrett.

**Methodology:** Julian Q. Kosciessa, Niels A. Kloosterman, Douglas D. Garrett.

**Project administration:** Julian Q. Kosciessa, Douglas D. Garrett.

**Resources:** Niels A. Kloosterman, Douglas D. Garrett.

**Software:** Julian Q. Kosciessa, Niels A. Kloosterman.

**Supervision:** Douglas D. Garrett.

**Validation:** Julian Q. Kosciessa.

**Visualization:** Julian Q. Kosciessa.

**Writing – original draft:** Julian Q. Kosciessa.

**Writing – review & editing:** Julian Q. Kosciessa, Niels A. Kloosterman, Douglas D. Garrett.

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
