## [Decision Letter · Decision Letter 0]

17 Feb 2020

Dear Mr. Kosciessa,

Thank you very much for submitting your manuscript "Standard multiscale entropy reflects spectral power at mismatched temporal scales: What’s signal irregularity got to do with it?" for consideration at PLOS Computational Biology.

As with all papers reviewed by the journal, your manuscript was reviewed by members of the editorial board and by several independent reviewers. The reviews (below this email), show a general appreciation of the methodology and attention to an important problem, as well as suggesting some possible issues and improvements.

Another important aspect is the relevance to the scope of the journal, which raised some concerns both with the reviewers and in editorial discussions. It would then be crucial to show an actual relevance in terms of better understanding of the biological mechanisms behind the signal properties. This might be difficult to show, and in that case, upon a resubmission with convincing answers to the other points, we would offer you immediate publication in PLOS One.

We cannot make any decision about publication until we have seen the revised manuscript and your response to the reviewers' comments. Your revised manuscript is also likely to be sent to reviewers for further evaluation.

Sincerely,

Daniele Marinazzo

Deputy Editor

PLOS Computational Biology

Daniele Marinazzo

Deputy Editor

PLOS Computational Biology

Reviewer's Responses to Questions

**Comments to the Authors:**

Reviewer #1: This technical paper investigates thoroughly, in simulations and with representative EEG data, two issue that affect the interpretability of multiscale entropy (MSE), a widely used tool for the quantification of the complexity of physiological time series performed at multiple time scales: the impact of one specific parameter (the “similarity bound”) on the assessment of complexity at long (coarse) time scales, and the impact of dominant low frequency oscillations on the assessment of complexity at short (fine) time scales.

The two issues are actually well known in the literature, and (as also partially acknowledged by the authors) corrective methodological approaches have been proposed to successfully deal with them (see specific comments below). This aspect should be better emphasized in the paper. Nevertheless, in spite of the limited methodological novelty and neurophysiological appeal of the results, the messages conveyed by the paper are important, because it is true that MSE is often misused or at least used not appropriately, and that inferences based on this measure must be made with full awareness of its meaning and limitations (which is often not the case in neuroscience and physiology applications). Therefore, I recommend acceptance of this paper after the content is simplified, in order to focus on the specific message, some interpretations are better placed in the context of the existing methodological literature, and the general claims are smoothed a bit, in order to better acknowledge that existing methodological solutions already make MSE a useful tool - provided that it is used consciously.

Specific:

- The effect of the signal variance (here reflected in the term “similarity bound”) on the original formulation of MSE is long known (e.g., from Valencia et al. [R1]). Here it is confirmed in simulations, and its effects are highlighted in the practical application studying age differences in resting state EEG. The proposed solution to compute sample entropy based on the variance of the signal after the change of scale (here denoted as “use of local similarity bound”) is also well established (it is for example used in linear parametric estimators of MSE (e.g., Faes et al. [R2])). The recommendation to abandon the original MSE formulation (line 678) is valid, but should be substantiated more clearly in terms of existing modifications of the MSE algorithm (like the refined MSE).

- Also the diffuse reflection of slow oscillations on entropy estimated even at fine time scales (i.e., a contribution of low frequencies to broadband signals) is expected, since at fine time scale the signal contains both low and high frequencies which can both contribute to the irregularity. In particular, the dominance of very low frequency components (where ‘very low’ relates to the sampling frequency and the length of the time series) introduces trends in the analyzed signal which can be assimilated to nonstationary behaviors; this precludes a proper evaluation of Sample Entropy, because stationarity is a formal prerequisite to the evaluation of entropy measures. This issue is thoroughly dealt with in recent works (e.g., Xiong et al. [R3]). Solutions have been proposed, either based on simple detrend of the time series via highpass filters in order to allow MSE to focus on short-term dynamics without the biasing effect of slow trends, or based on incorporating the long-range correlations into parametric models for the estimation of MSE (Faes et al. [R4]) in order to describe both long- and short-term dynamics in MSE analysis. These aspects should be better emphasized in the paper, in order both to better place the work in the context of the existing literature and to substantiate the recommendation of using spectral filters to deal with the second issue highlighted in this work.

- The paper correctly questions the interpretation often given to MSE that it reflects the complexity of the analyzed signal observed at specific time scales. In doing this, however, it should be stated more explicitly that the mismatch between specific temporal scales and MSE values is a defining feature of this measure, which distinguishes it from simple spectral analysis. The use of lowpass filters encompassing several temporal scales is indeed important in order to allow the measure to capture complex (possibly nonlinear) behaviors like the superposition of rhythms and at different frequency and their weaker or stronger interaction. Therefore, it should emerge more clearly from the paper that, while it is incorrect to interpret MSE as a scale-specific measure, the “scale” focused by this measure is actually an upper limit of the range of scales analyzed (i.e., at “scale 1”, all signal frequencies are considered, and at scale tau all frequencies lower than fs/(2+tau) are considered – where fs is the sampling frequency).

Related to the comment above, two issues should be also considered:

(i) not only it is wrong to assume that MSE reflects a specific time scale, but also it makes little sense to look scale-specific MSE values through narrowband filters; the authors correctly show that narrowband filters reconcile MSE with the spectral content (e.g. in Fig. 4E), but they should acknowledge that in that case MSE itself becomes of little use, since it merely reflects the presence of specific oscillations and thus it adds little or no information to much simpler spectral analysis (the evaluation of nonlinear effects and cross-frequency interactions is precluded by the narrowband filter).

(ii) the message that MSE is of little use because it reflects spectral content at mismatched temporal scales (given first of all in the title of the article) is misleading and should be tuned down. While it is true that a the balance among spectral content at different frequencies has an effect on the MSE value, such a balance is one of the factors that determines signal complexity (e.g., white noise is highly complex, an oscillation embedded in noise is less complex, the presence of another oscillation at different frequency increases complexity, the complexity increases with the bandwidth of stochastic oscillations, and so on)

- The paper is very well written, but it is likely too long compared to the message that is given. An effort to focus on the main message would be helpful to better convey it.

[R1] Valencia, J. F., et al. (2009). Refined multiscale entropy: Application to 24-h holter recordings of heart period variability in healthy and aortic stenosis subjects. IEEE Transactions on Biomedical Engineering, 56(9), 2202-2213.

[R2] Faes, L., et al. (2017). Efficient computation of multiscale entropy over short biomedical time series based on linear state-space models. Complexity, 1768264.

[R3] Xiong, W., et al. (2017). Entropy measures, entropy estimators, and their performance in quantifying complex dynamics: Effects of artifacts, nonstationarity, and long-range correlations. Physical Review E, 95(6), 062114.

[R4] Faes, L., et al. (2019). Multiscale information storage of linear long-range correlated stochastic processes. Physical Review E, 99(3), 032115.

Reviewer #2: Overview

This manuscript, by Kosciessa and colleagues, investigates methodological properties of applying multiscale entropy (MSE) to neural field data, such as EEG. The research question is broadly, whether MSE actually measures signal irregularity across scales, in the way that it is typically interpreted to do. More specifically, this investigation centres on two key methodological points about how typical approaches compute and analyze the 'scales' that are foundation of the 'multi-scale' approach, specifically:

whether using a global similarity bound biases measures across scales, since the coarse-graining (or analogous) procedure reduces the variance as one restricts the data to different scales, and so a globally computed similarity bound effects each scale differently and thus conflates signal regularity and signal variance; and

whether the coarse graining procedure is specific to presumed scales, or whether, for example, scales that include, but are not limited to, high frequency activity capture entropy measures that are specific to high frequency activity, or whether the low frequency activity also impacts the measures and thus conflates the typical interpretations of measures at particular scales.

The authors demonstrate, in a series of simulations, that these two methodological issues are indeed problems, such that global similarity bounds and broadband data lead to biases that violate how the resulting measures are typically interpreted. In analyses of EEG data, they further demonstrate how this is the case, showing how patterns of MSE relate to different similarity bounds and coarse graining approaches. These analyses also demonstrate how MSE measures relate to power. Through this work, they propose and investigate solutions to these problems: notably to recompute similarity bounds specific to the variance at each scale, and to use filters to isolate data more specifically to desired scales.

Overall, I commend the authors on a rigorous and methodologically minded paper, that demonstrates methodological insight into a relatively common method in the field, demonstrated through what I find to be a compelling and well chosen empirical analysis. I was very impressed by the methodological rigour of the work, the open nature of the code and data, and the clear demonstration of how these methodological points work, and what they mean. This seems to me to be important work for the field, and impactful work in terms of thinking of how to interpret many studies that have employed these methods.

Overall, I did not find any major issues with the overall framing, logic or main results of this paper. I also find it to be well written. My review is therefore focused on a what I consider to be some minor revisions that include some clarifications regarding their proposed adaptations MSE, a possible small extension to the simulations, and some possible tweaks to the figures.

Comments:

1) Bandpass Filtering

My main conceptual question / comment is about the use, interpretation and recommendation for the bandpass filtered version of the entropy. I like the idea of this approach, and the justification of how it addresses the issues raised, and is a design that actually specifies scales precisely. However, it still seems slightly unclear what it captures, and if (or to what extent) this is a) the same as the typical conceptualization of MSE and b) distinct from other measures (such as power and burst analysis).

In particular, the demonstration that alpha and beta differ in bandpass MSE between groups is very interesting. However, given that the alpha does relate to power, and the beta does relate to the burst analysis, I found it unclear if the there is a conclusion as to whether this MSE approach adds information over and above a combined power & burst analysis of rhythmic activity. The implication in the results appears to be that the difference in beta is proposed to be related to different bursting statistics of rhythmic activity between groups. If this is the appropriate interpretation, it seems to me that a) it is not entirely clear to me that tracking burst properties (such as probably and duration of bursts) of rhythmic activity is quite the same idea as MSE has been proposed to relate to - of signal irregularity (seemingly, more broadly). This could perhaps just be a conceptual issue, as burst dynamics are clearly an aspect of signal (ir-)regularity, but my impression is that entropy measures are more typically thought of as representing the 'eracticness' of time series, and not necessarily the propensity for bursts of rhythmicity. It is also somewhat unclear to me if this MSE analysis offers something more than could be gleaned from a power and burst analysis of these bands.

Specifically, I would appreciate a comment in the manuscript that addresses 1) whether this MSE measure adds something more than would be gleaned from a power and burst analysis and 2) whether, if one does try this, the results should be interpreted somewhat differently than how 'original MSE' has typically been interpreted and discussed.

Some other questions came to mind about this narrowband MSE measure:

It seems very likely the estimate of MSE calculated from bandpassed data would be dependent on the width of the bandpass filters applied. Is this the case? And if so, what does this relation mean, in practice, for computing this measure, and how should one choose bandwidths?

Narrow bandpass filters (and the bandpass filters applied here seem quite narrow, at the lower frequencies) enforce sinuisoidality onto the filtered data. Narrowband filtering then removes what may be considered interesting waveform shape properties that one might have considered could be an interesting driver of entropy differences. This seems potentially related to the notion that this form of MSE may not capture some desired notions of signal irregularity. The question is then: how does the narrowband filtering effect waveform shape in a way that might affect entropy measures?

Should one consider potential differences in centre frequency if one is comparing between groups? The two groups in this analysis differ in age in a manner that would be expected to demonstrate a difference in peak alpha frequency. Given this, could or should one consider aligning compared bands to individual centre frequencies? If not, to what extent might difference in entropy simply relate to differences in centre frequency?

To be clear, I think that a full investigation of all the properties of the bandpass filter version of MSE could quickly become large, and that is out of scope of this particular manuscript. I don't consider that these 'other questions' must all be addressed or answered in this manuscript. They are suggested to motivate that it is perhaps worth adding a quick discussion point noting that there are open questions (and that there may be some limitations) to the narrowband MSE approach, and that further work is needed on this.

2) Simulating Variable 1/f

If there is one asymmetry between what is done in the simulations and what is presented in the real data, I would say it is that a key point made in the EEG data is how the difference in 1/f backgrounds of the data is a key factor in affecting MSE measures, and yet this point is not demonstrated in the simulated data. All of the simulations use a single value for x in 1/f^x. While I think the inference and interpretation of the 1/f effects are clear, if it is relatively straight-forward to compute and add, a set of simulations across different 1/f components of the simulated data (perhaps just 2 different values for x, representing each group) might be a nice addition. This would clearly show, in the simulated data, that this effect can be isolated to changes in the 1/f structure of the data.

3) Values for the PSD Slopes

A minor comment, for clarification, is regarding the actual values for the PSD slopes that are plotted, for example in Figure 6 D1 and in Figure S6. The actual values are very small, and not typical of values for PSD slopes, which are typically closer to values of -1 (equivalent to 1/f^1, or pink noise). A 'back of the envelope' estimated calculation of the slope of the spectrum plotted in Figure 6 A2 is also more consistent with this value or around -1. Have the potted numbers been transformed in some way? I found no mention in the methods that would explain this, and so some clarification of the values plotted would be useful. Perhaps related, it is unclear why the words 'minus 0' appear in the title of the panel D1.

4) Figure Tweaks

Finally, I have is actually a collection of notes and small suggestions for the figures. There are a lot of figures which nicely present a lot of data and findings, but I feel like some small updates would help make the figures a little more accessible to the new reader.

Figure 1

In 1B, the 'similarity bound' is indicated as '(r x SD)', where SD presumably standards for standard deviation. However, in the figure legend, similarity bounds are defined as 'r * 0.5 of signal variance'. It would be clearer if one form of the definition was used, which based on the other figures, I presume should be the one using standard deviation. It would be useful to define 'SD' here.

Figure 2

It's a small aesthetic nitpick: but the y-axis labels for A & B are not aligned

Figure 4

I think it's a mistake that the legend refers to a 'blue-to-red' line gradient?

Fig 6:

I initially found it slightly unclear on how to read and interpret the contrasts and topographies. If the paper is to be organized with the methods at the end (as current), I think it would be useful to note in the figure legend a brief note on what the p-values are that are plotted (these are presumably the results of the significance of the cluster-level statistic?)

The legend suggests asterisks are used to indicate significance, but this does not appear to be used in the plot?

I initially found the labelling to be unclear and wasn't sure what the 'B1', 'C1', etc labels were in panels 'A1' and 'A2'. Assuming these refer to the other sub-panels (and seem to motivate the labelling) briefly noting what these refer to in the legend could be useful.

Is A1 the 'Original' measure of entropy? This should indicated in the title and/or legend.

Figure 7:

I'm unclear on why the x-scale for Row 1 appears to be different from how it was plotted in Figure 6A1. Is what is plotted in Figure 6A1 not a copy of one of the plots in Figure 7 Row 1? It would be easier to highlight this if the scale were the same. Perhaps there is a reason they are different, but this could be noted, as otherwise it initially make the plotted measures between figure seem more different than they actually are.

Could you label the rows? In particular, I found it somewhat unclear as to what comparisons rows 2 and 4 were exactly, and especially row 4 (I believe it is the difference of similarity bounds, between groups, across channels?). This could be indicated more clearly in the figure legend.

It would be useful to you indicate (probably in the legend) in which direction the comparison was made, so that the direction of the t-values can be interpreted with respect to groups.

Figure 8:

The title in C says that entropy difference is <8 Hz, but the figure legend reports it's < 6 Hz.

Figure 9:

The labels for the lines in B are not clearly differentiated. The long dash and full solid labels look the same in the way they are plotted in the legend.

Figure 10:

Are all the topographies on the same color bar? The color bar is not consistently plotted, and if one is to be used across all topographies, this could be noted in the legend.

For G & H, are these collapsed across age groups? This would be useful to clarify.

Figure 11:

A stylistic suggestion is that I don't think you need the 'higher frequencies' arrow twice (at both the top and the bottom).

I initially found it somewhat unclear as to how to interpret the dashed coloured arrows. I'm not entirely sure if there is a change to be made, though perhaps they could be briefly mentioned in the figure legend, for clarification?

Reviewer #3: This is a comprehensive and thought-provoking assessment of the relation between spectral power (SP) and multiscale entropy (MSE), both of which have been used to characterize changes in brain dynamics across a wide range of conditions (mainly from EEG data). One of the perpetual challenges with MSE is the attempt to relate the scales for downsampling to something akin to frequency. This has some support in that there does appear to be some similarity in scale differences in MSE and frequency differences in SP. For example, the authors show age-related differences in MSE (higher fine-scale, lower coarse-scale) map to SP changes with higher power in faster frequencies and lower at slower frequencies - similar to what we (McIntosh) and others have published. While it is tempting to link these two metrics, I think most agree that it is not so straightforward (figure 11 sort of suggests this).

The first point that is emphasized is that the "original" MSE algorithm does not change the similarity criteria with successive downsampling of the signal. This appears to introduce a bias at coarse scales. This is a concern and definitely needs to be considered for MSE applications. The authors quite convincingly show that this can affect interpretations of group differences (Fig 7).

The second point, if I understand correctly, is that a direct mapping between scale and frequency is difficult because of cross-spectral dependencies. This isn't too surprising and in my read of the extant literature, I think most would agree. The Courtiol paper, for example, does make this point, and most of the work we did notes the similarities in MSE and SP, but also that they are sensitive to different aspects of brain signals. For example, nonlinearities (e.g., cross-spectral dependencies) do affect MSE, but by definition do not affect SP. Indeed, the surrogate analysis the authors mention (which was done in the McIntosh et al, 2008 paper), shows this quite nicely. Also some other work (Bruce, Bruce, Vennelaganti, 2009) suggested that better predictor of sample entropy was found to be related to the power ratio of higher to lower frequencies:(alpha + beta)/(delta + theta). Moreover, the direct link between distance (local vs distributed), SP and MSE is also not straightforward. For example, while the age-effects reported in the present paper and by others do replicate nicely, there does appear to be a age-related change in local entropy (higher in ageing) and mutual information (lower in ageing), which maps to coherence estimates (McIntosh et al, Cereb Cortex 2014). These "connectivity" effects, however, are not constrained to a particular scale or frequency, which emphasizes the inherent nonlinearity in these effects. The Courtiol et al work also notes this sensitivity to nonlinearities.

Thus, I am left wondering if the second message is as conclusive as the authors seem to suggest. It's complicated, no doubt, as the Venn diagram in Figure 11 suggests. Which takes me to the main point. I get the impression that the authors advocate SP as the 'gold standard' on which to map the MSE effects, but I am not sure this is a valid. The simulation used to support this changes power at the alpha frequency in an 1/f signal, which makes a very strong assumption on the underlying biological processes. Thus, insofar as the brain signals actually do this -- add power to a restricted frequency band -- the simulation is useful. But, if there are changes in the brain signal that span frequencies (e.g., cross-spectral dependencies), then the spectral filters won't really help and may in fact obscure things. I suggest the authors actually do the surrogate analyses they propose as suggestion (d) on page 32 to really disentangle this. Otherwise, I am concerned that someone who reads this will come away thinking that measuring SP is really all we need to do, which I don't think is the message the authors want to convey.

Reviewer #4: This is a well-done paper. The authors should be congratulated for their thorough and didactic exposition of an intricate problem. However, the study left me unconvinced that, at least in its current state, it has a place in a leading computational neuroscience/biology journal. I will list my concerns in more detail below, but will have to leave it to the editors to weigh my comments against the broader context of the scope of their journal.

The paper starts out with multi-scale entropy (MSE) being a key measure, “increasingly”, "often", "commonly" applied in a specific way, which the authors go on to explain in admirable detail (i.e., applying a broad-band variance scaling). The authors then go on to show how this creates a severe band-specific spectral power confound. Using cross-sectional data from young and older brains they go on to argue that conclusions on entropy/variability changes over the life-span could as well be confounded by power. They end on a set of cook-book-style recommendations. I have not much to quibble with in the way they set out this argument.

The overriding impression, however, was that of a data analysis tutorial or a methods paper. The implications for our understanding of neural functioning and/or neuro-cognitive computation did not become clear.

From the outset, I had been expecting a closer link to the neurobiology of (multi-scale) entropy, and I do think that such a link is necessary to lift the current paper, and its potential impact, beyond the ranks of a methodological comment.

Not least, the mixture of simulations and data re-analysis left me furthermore unclear what I am to take from this study. For a hardcore signal analysis/digital signal processing audience, some of the treatment might be too shallow, while for the applied neuroscientist the particular example of age-related changes in spectral slope might be too specific to fully take away what the consequences for future MSE applications might be.

Should a revision be invited, I would suggest to focus the paper more on what we learn about age-related change in the spectral composition / E:I imbalance in the aging brain and would suggest to tone down the overly specific, cook-book-like recommendations on how to analyse MSE instead.

Minor: SD and variance are used a bit too interchangeably, especially aroudn Fig. 1/caption thereof.

**Have all data underlying the figures and results presented in the manuscript been provided?**

Reviewer #1: Yes

Reviewer #2: Yes

Reviewer #3: Yes

Reviewer #4: Yes

PLOS authors have the option to publish the peer review history of their article (what does this mean?). If published, this will include your full peer review and any attached files.

Reviewer #1: Yes: Luca Faes

Reviewer #2: Yes: Thomas Donoghue

Reviewer #3: Yes: Anthony Randal McIntosh

Reviewer #4: No
---

## [Decision Letter · Decision Letter 1]

18 Apr 2020

Dear Mr. Kosciessa,

We are pleased to inform you that your manuscript 'Standard multiscale entropy reflects neural dynamics at mismatched temporal scales: What’s signal irregularity got to do with it?' has been provisionally accepted for publication in PLOS Computational Biology.

I just have one request, which will not influence the decision, but that I would like you to address: in figure 11 please consider removing bar plots and error bars and substituting them with scatter plots, or at least boxplots with quantiles,in order to have an idea on the distribution of data and of the presence of possible outliers, which ultimately will justify the choice of the statistical test used.

Also, p values come from a uniform distribution, and discretizing it in levels marked by asterisks, even if it's a common practice, is in my opinion misleading. Just report p values with effect sizes. Significance is a subjective measure anyway.

Best regards,

Daniele Marinazzo

Deputy Editor

PLOS Computational Biology

Reviewer's Responses to Questions

**Comments to the Authors:**

Reviewer #1: The Authors have replied exhaustively and satisfactorily to all comments raised in my first revision. I suggest acceptance of the paper, which represents an important contribution that will be hopefully taken into consideration by the high number of researchers employing Multiscale Entropy and related tools for the analysis of neurobiological signals.

Reviewer #3: The authors have done a tremendous job of address my concerns (and those of the other reviewers in my opinion). I commend them on good work

**Have all data underlying the figures and results presented in the manuscript been provided?**

Reviewer #1: Yes

Reviewer #3: Yes

PLOS authors have the option to publish the peer review history of their article (what does this mean?). If published, this will include your full peer review and any attached files.

Reviewer #1: Yes: Luca Faes

Reviewer #3: Yes: AR McIntosh

---

## [Editor Report · Acceptance letter]

28 Apr 2020

PCOMPBIOL-D-20-00041R1 

Standard multiscale entropy reflects neural dynamics at mismatched temporal scales: What’s signal irregularity got to do with it?

Dear Dr Kosciessa,

I am pleased to inform you that your manuscript has been formally accepted for publication in PLOS Computational Biology. Your manuscript is now with our production department and you will be notified of the publication date in due course.

With kind regards,

Laura Mallard
